# Towards Federated Learning of Deep Graph Neural Networks

## Abstract

Graph neural networks (GNNs) learn node representations by recursively aggregating neighborhood information on graph data. However, in the federated setting, data samples (nodes) located in different clients may be connected to each other, leading to huge information loss to the training method. Existing federated graph learning frameworks solve such a problem by generating missing neighbors or sending information across clients directly. None are suitable for training deep GNNs, which require a more expansive receptive field and higher communication costs. In this work, we introduce a novel framework named Fed$^2$GNN for federated graph learning of deep GNNs via reconstructing neighborhood information of nodes. Specifically, we design a graph structure named *rooted tree*. The node embedding obtained by encoding on the rooted tree is the same as that obtained by encoding on the induced subgraph surrounding the node, which allows us to reconstruct the neighborhood information by building the rooted tree of the node. An encoder-decoder framework is then proposed, wherein we first encode missing neighbor information and then decode it to build the rooted tree. Extensive experiments on real-world network datasets show the effectiveness of our framework for training deep GNNs while also achieving better performance for training shadow GNN models[1].

## 1 Introduction

Recently, Graph Neural Networks (GNNs) have attracted significant attention due to their powerful ability for representation learning of graph-structured data (Hamilton et al., 2017a; Kipf & Welling, 2017; Hamilton et al., 2017b). Generally speaking, it adopts a recursive neighborhood aggregation (or message passing) scheme to learn node representations by considering the node features and graph topology information together (Xu et al., 2018). After $k$ iterations of aggregation, a node captures the information within the node's $k$-hop neighborhood.

Similar to learning tasks of other domains, training a well-performed GNN model requires its training data to be not only sufficiently large but also heterogeneous for better generalization of the model. However, in reality, heterogeneous data are often separately stored in different clients and cannot be shared due to policies and privacy concerns. To that end, recent works have proposed federated training of GNNs (Zhang et al., 2021; Peng et al., 2021; Yao & Joe-Wong, 2022; Chen et al., 2022). They typically consider a framework wherein each client iteratively updates node representations with a semi-supervised model on its local graph; the models are then aggregated at a central server. The main challenge is that data samples (nodes) located in different clients may be connected to each other. Hence, it is non-trivial to consider the connected nodes (i.e., neighbor nodes) located in other clients when applying node updates. Although existing works focus on recovering missing neighborhood information for nodes, they either only consider immediate neighbors (Zhang et al., 2021; Peng et al., 2021) or require communication costs to increase exponentially as the neighbors' distance increases (Yao & Joe-Wong, 2022; Chen et al., 2022). None of them are suitable for training deeper GNN models, which require a more expansive receptive field and have been shown to be beneficial for representation learning for graph-structured data (Li et al., 2019; Liu et al., 2020;

---

[1]Code available at `https://www.dropbox.com/s/unizcyixsmip0je/Fed%5E2GNN.zip?dl=0`

Zhou et al., 2020a). For GNNs, the receptive field of a node representation is its entire neighborhood. Moreover, (Yao & Joe-Wong, 2022) also requires calculating the weighted matrix in advance, which is not available in practice.

In this work, we aim to fundamentally address the above limitations of existing federated graph learning methods by proposing a novel framework named Fed$^2$GNN. The key idea lies in designing a principled approach to reconstructing the neighborhood information of nodes that considers both structure-based (i.e., graph topology) and feature-based information (i.e., node features). For the structure-based information, we propose a novel graph structure named *rooted tree*, which has a more regular structure than the original structure of the node neighborhood. More importantly, the node embedding obtained by encoding on the rooted tree is the same as that obtained by encoding on the node's ego-graph (i.e., the induced subgraph surrounding the node). Such a property allows us to easily reconstruct the structure-based information by building the rooted tree of the node.

For the feature-based information, since the structure of the node neighborhood changes, we aim to generate features of the nodes in the rooted tree. Inspired by the structure of the rooted tree, we design a protocol wherein clients recursively transmit information across each other. The data transmitted in the $k$-th round correspond to nodes in the $k+1$-th layer of the rooted tree. Furthermore, we utilize the encoder-decoder framework to reduce the communication costs such that it grows only linearly as the number of iterations increases. In more detail, each client first encodes the information and sends the output to other clients. Other clients build the rooted tree by decoding the received information. By merging all trees into the local graph (with the rooted node as an anchor), each client obtains a complete graph on which applying graph representation learning has limited information loss. In summary, we make the following contributions:

- We introduce Fed$^2$GNN, an framework for federated training of GNNs to solve node-level prediction tasks. We achieve such a goal by devising a principled approach to reconstructing missing neighborhood information that considers both structure-based and feature-based information.

- To reconstruct the structure-based information, we propose a novel graph structure named *rooted tree*, which is easier to construct than the original irregular structures of the node neighborhood. More importantly, the node embedding obtained by encoding on the rooted tree is the same as that obtained by encoding on the node's ego-graph.

- To reconstruct the feature-based information, we propose an encoder-decoder framework to reduce communication costs while having limited information loss.

- We conduct extensive experiments to verify the utility of Fed$^2$GNN. The results show that it is effective for training deep GNNs while achieving better performance for training shadow GNN models.

We outline related works in Section 2 before introducing the problem statement of federated graph learning in Section 3. We then introduce Fed$^2$GNN in Section 4, wherein we first introduce the structure of the rooted tree and then presents the neighborhood reconstruction process. We analyze its performance experimentally in Section 5 and concluding in Section 6.

## 2 RELATED WORKS

**Graph Neural Networks (GNNs)** learn a representation for each node in the graph using a set of stacked graph convolution layers. Each layer gets an initial vector for each node and outputs a new embedding vector by aggregating vectors of neighbor nodes followed by a non-linear transform. After $k$ aggregations, the source of information encoded in the representation of a node essentially comes from its $k$-hop neighborhood. Following the above framework, which is usually called message passing, several GNN models have been proposed, such as GCN (Kipf & Welling, 2017), GraphSAGE (Hamilton et al., 2017b), GAT (Velickovic et al., 2018), and so on. However, unlike the learning tasks in other domains, simply stacking graph convolution layers usually suffer from an over-smoothing issue, leading to even worse performance. Surprisingly, with the research on the above issues, several works (Liu et al., 2020; Li et al., 2019; Zhou et al., 2020b) propose effective deep GNNs and obtain better performance on graph learning tasks. Its excellent performance suggests its great potential for federated learning on distributed subgraph data.

**Federated Learning (FL)** is a privacy computing technology that enables collaborative machine learning without exposing private data. It was first proposed by (McMahan et al., 2017)'s FedAvg, which allows clients to collaboratively train a common model while each of them only possesses a local dataset. During training, each client periodically uploads the local update to the server. The server then aggregates the updates to a global model and distributes the model to clients for further training. Recently, personalized federated learning (Finn et al., 2017; Li et al., 2020), which aims to learn heterogeneous models for different local tasks, also attracted attention from the community. However, our paper focuses on learning a global model from multiple clients for a common task. Hence, we mainly borrow the idea of FedAvg to train GNNs collaboratively.

**Federated Graph Learning** aims to solve the graph learning problem on distributed subgraph data. Recent researchers have made progress on federated graph learning, and several frameworks have been proposed (Fu et al., 2022; Liu & Yu, 2022). (Zhang et al., 2021) proposed a framework named FedSage+, which utilizes a neighbor generator to recover cross-client neighbors of subgraphs. However, they only consider immediate neighbors and can not fully recover the cross-client information. FedGraph (Chen et al., 2022) develops a cross-client graph convolution operation to enable embedding sharing among clients, which requires sending updated embedding in every iteration during model training. FedGCN (Yao & Joe-Wong, 2022) solves such a problem by proposing to transmit the aggregated features of the cross-subgraph neighbor nodes directly. However, it assumes that the weighted matrix is calculated in advance and requires communication costs to increase exponentially as the number of layers increases. Baek et al., (Baek et al., 2022) propose a personalized federated subgraph learning method, mainly focusing on heterogeneous subgraphs. In more detail, it develops an aggregation method based on similarity among clients. However, it does not consider the missing neighbor information. In summary, none of the existing works are suitable for federated learning for deep GNNs; we are the first to achieve such a goal.

## 3 PROBLEM STATEMENT.

Given an attributed non-directed graph $\mathcal{G} = (\mathcal{V}, \mathcal{E}, \mathcal{X})$, where $\mathcal{V}$ is the vertex set, $\mathcal{E}$ is the edge set, and $\mathcal{X}$ include the node features. Each node has a feature $x_v \in \mathcal{X}$ with a corresponding label $y_i$ for the downstream task, e.g., node classification. We have $x_v \in \mathbb{R}^{d_x}$.

In the FL system, we assume there exist a central server $S$ and $M$ clients $\mathcal{P}_p, p \in [M]$, where $[M]$ represents $\{1, ..., M\}$ for any $M \in \mathbb{N}^+$. The server only maintains a graph learning model with no graph data stored, and each client holding a subgraph $\mathcal{G}_p = (\mathcal{V}_p, \mathcal{E}_p, \mathcal{X}_p)$, where $\mathcal{V}_p, \mathcal{E}_p, \mathcal{X}_p$ are subset of $\mathcal{V}, \mathcal{E}, \mathcal{X}$ respectively. We assume no overlapping nodes shared across clients, namely $\mathcal{V}_p \cap \mathcal{V}_q = \emptyset, \forall p, q \in [M], p \neq q$, and all subset constitute the full set, i.e.,$\bigcup^p \mathcal{V}_p = \mathcal{V}$. Given a node $v \in \mathcal{V}$, we define its $k$-hop neighborhood as the nodes which have shortest paths of length $k$ from $v$. Let $c(v)$ denote the index of the client that contains node $v$ and $\mathcal{N}_v$ denote the 1-hop neighborhood of $v$. For any node $v \in \mathcal{V}$, we assume $\mathcal{P}_{c(v)}$ knows all $v$'s 1-hop neighbors even they are distributed across different clients. The set of neighborhoods contained in the same client as $v$ is denoted as $\mathcal{N}_v^p$, and the set of other neighborhoods is denoted as $\mathcal{N}_v^{cp} := \mathcal{N}_v \backslash \mathcal{N}_v^p$. We name $\mathcal{P}_{c(v)}$ as the **active client** and other clients $\{\mathcal{P}_{c(j)} | j \in \mathcal{N}_v^{cp}\}$ as **passive clients** of $v$ for ease of expression.

## 4 METHODS

In this section, we propose a novel framework for federated learning of deep GNNs (Fed$^2$GNN). The framework relies on a principled approach to reconstructing neighborhood information, with the core idea of building rooted trees for all nodes. After constructing all trees, we can get complete local graphs, on which applying graph representation learning has no neighborhood information loss. The FedAvg (McMahan et al., 2017) algorithm is then adopted to train a GNN on complete local graphs. Next, we start with designing a rooted tree of a node and then show how to build the tree with an encoder-decoder framework.

### 4.1 DESIGN OF THE ROOTED TREE

Given a $K$-layer GNN model, it learns node representation by recursively aggregating neighborhood information around the node. After $K$ times aggregation, it could encode information from the

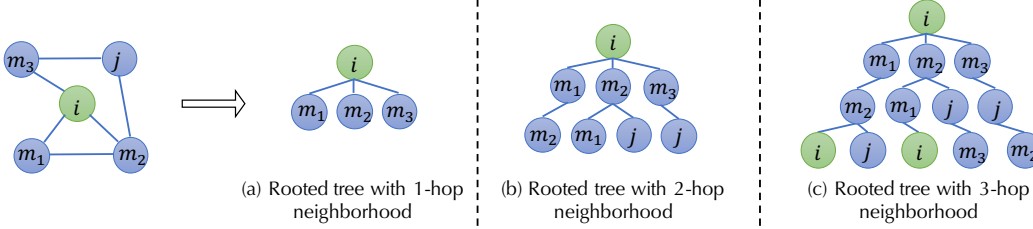

Figure 1: Examples of constructing rooted trees with different neighborhood for node $i$.

$k$-hop subgraph of the node, wherein the subgraph is often called the ego-graph. In order to reconstruct the missing neighborhood information for each node, the rooted tree is designed by following two principles: (1) it could fully preserve the neighborhood information, i.e., the node embedding obtained by encoding on the rooted tree is the same as that obtained by encoding on the node's ego-graph, and (2) it has an easy-to-build structure. To that end, we design the rooted tree by unfolding the ego-graph for each node.

An example of constructing rooted trees with 1,2,3-hop neighborhoods is depicted in Figure 1. Specifically, for a node $i \in \mathcal{G}$, we first set $i$ as the rooted node. We then can set $i$'s 1-hop neighbors $m \in \mathcal{N}_i$ as nodes in the second layer of the rooted tree (Figure 1 (a)). The third layer, intuitively, can be constructed using $m$'s neighbors for each $m \in \mathcal{N}_i$. However, note that $i$ is also included in $\mathcal{N}_m$ and connected to it as the father note of $m$. Hence, we exclude $i$ and connect the remaining nodes in $\mathcal{N}_m$ to $m$ to further construct the rooted tree (Figure 1 (b)). Following the above procedure, we construct the $k+1$-th ($k > 1$) layer by connecting the neighbors of nodes in the $k$-th layer to itself except for its father node. Finally, for a $K$-layer GNN model, we construct a rooted tree with $K+1$ layers.

**Proposition 1.** *Given a $K$-layer GNN model, for any node $i$ and its corresponding ego-graph, we can construct a $K+1$-layer rooted tree following the above procedure such that $i$'s embedding obtained by encoding on the rooted tree is the same as that obtained by encoding on its ego-graph.*

It's worth noting that the rooted tree is actually an undirected graph. For any node on the tree, it treats its father node and children nodes as one-hop neighbors and recursively aggregates their information to update its embedding. Meanwhile, the nodes in the $k$-th layer of the rooted tree have the complete neighborhood information of $K+1-k$ hops, and the rooted node at the first layer have the complete $K$-hop neighborhood information. Thus, for a K-layer GNN model, a rooted node has the same embedding as obtained by encoding on its $K$-hop ego-graph. Meanwhile, although the degree of nodes in the last layer changes, which affects the convolution results of nodes at $K$-th layer to models like GCN (Kipf & Welling, 2017), we can manipulate the edge weight to guarantee the correctness of the convolution process. Details are presented in Appendix A.

In practice, we aggregate the leaf nodes into a single node. It does not affect the encoding process when applying the sum function to aggregate features of neighbor nodes. Meanwhile, we only construct the rooted tree of missing neighborhood information. For instance, assume that nodes $i, m_1, m_3, j$ in Figure 1 are located in the same client while node $m_2$ locates in a different client. We can construct the rooted tree of $i$ by only considering the $m_2$ branch in the second layer. Although $m_2$ also acts as $i$'s 2-hop neighbor node, affecting $i$'s encoding result through the path $m_2 - m_1 - i$, which does not exist in $i$'s rooted tree, we solve this problem by merging the rooted trees of $i$ and $m_1$ into the original subgraph, wherein the path $m_2 - m_1$ exists in $m_1$'s rooted tree. Finally, by merging rooted trees of all nodes into the local graph (with the rooted node as an anchor), we obtain a complete graph.

## 4.2 NEIGHBORHOOD RECONSTRUCTION

To construct the rooted tree of $K+1$ layers, we have clients recursively transmit information across each other for $K$ iterations. Meanwhile, an encoder-decoder framework is further used to reduce communication costs. An visual illustration of the information transmitting process across clients is presented in Figure 2. For ease of presentation, we start from 1-hop and 2-hop neighborhood reconstruction and then generalize to $K$-hop neighborhood reconstruction.

**One-hop neighborhood Reconstruction.** Constructing a rooted tree with 1-hop neighborhood in-

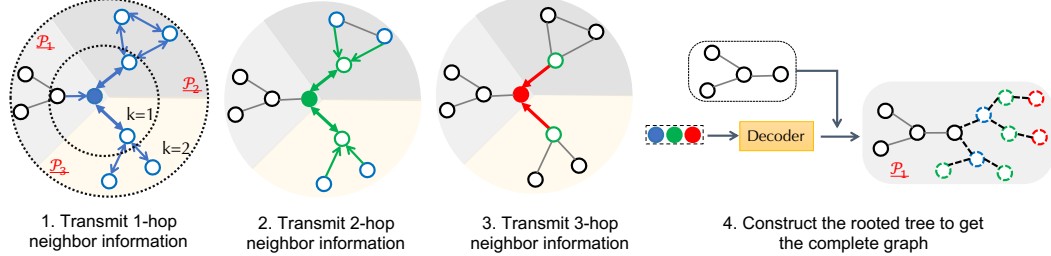

1. Transmit 1-hop
neighbor information

2. Transmit 2-hop
neighbor information

3. Transmit 3-hop
neighbor information

4. Construct the rooted tree to get
the complete graph

Figure 2: Visual illustration of the information transmitting process across clients and the approach to get the complete graph by constructing a rooted tree. The information transmitted between clients is encoded and represented by the bold arrow.

formation only requires the features of 1-hop neighbors of the rooted node. According to the expression of a 1-layer GNN model presented below,

$$\hat{y}_i = \sigma(\sum_{m \in \mathcal{N}_i \cup \{i\}} x_m W^{(1)}) = \sigma((\sum_{m \in \mathcal{N}_i^p \cup \{i\}} x_m + \sum_{m \in \mathcal{N}_i^{cp}} x_m) W^{(1)}). \tag{1}$$

It is sufficient for the passive client $\mathcal{P}_z$ send $\sum_{m \in \mathcal{N}_i^{cp}} \mathbb{1}_z[c(m)] x_m$ to the active client, where the indicator $\mathbb{1}_z[c(m)]$ is 1 if $z = c(m)$ and zero otherwise. The active client then sums all received information to get $f_i^1 := \sum_{m \in \mathcal{N}_i^{cp}} x_m$ and treats it as a pseudo node to construct the rooted tree. We treat the sum function as an encoder and do not require a decoder here. The same idea is also presented in (Yao & Joe-Wong, 2022). However, it assumes that the weight matrix is calculated in advance, which is unavailable when training GCN (Kipf & Welling, 2017) models in practice since the degrees of neighbor nodes in other clients are usually unknown.

**Two-hop neighborhood Reconstruction.** Constructing the rooted tree of 2-hop neighborhood information is a continuation of the process of 1-hop neighborhood Reconstruction. Specifically, for all $i \in \mathcal{G}$, we assume $\mathcal{P}_{c(i)}$ knows $f_i^1$ already.

Our intuition lies in the expression of a 2-layer GNN.

$$\hat{y}_i = \sigma(\sum_{m \in \mathcal{N}_i \cup \{i\}} \sigma(\sum_{j \in \mathcal{N}_m \cup \{m\}} x_j W^{(1)}) W^{(2)})$$

$$= \sigma((\sum_{m \in \mathcal{N}_i^p \cup \{i\}} \sigma((\sum_{j \in \mathcal{N}_m^p \cup \{m\}} x_j + \sum_{j \in \mathcal{N}_m^{cp}} x_j) W^{(1)}) + \sum_{m \in \mathcal{N}_i^{cp}} \sigma(\sum_{j \in \mathcal{N}_m \cup \{m\}} x_j W^{(1)})) W^{(2)})$$

$$\tag{2}$$

Although three items are missing to get $\hat{y}_i$, including the missing one-hop information $\sum_{j \in \mathcal{N}_i^{cp}} x_j$ as well as the missing two-hop information $\forall m \in \mathcal{N}_i^p, \sum_{j \in \mathcal{N}_m^{cp}} x_j$ and $\forall m \in \mathcal{N}_i^{cp}, \sum_{j \in \mathcal{N}_m \cup \{m\}} x_j$, only the third item remains unknown for $P_{c(i)}$ after constructing the 1-hop neighborhood for all nodes. To that end, we have passive clients encode the missing information first and send the result to the active client. Specifically, for a passive client $\mathcal{P}_z$, who owes a subset $\mathcal{N}_i^{cp-z} \subset \mathcal{N}_i^{cp}$, it calculate $q_z^2 := \sum_{m \in \mathcal{N}_i^{cp-z}} \mathbb{1}_z[c(m)] \sum_{j \in \mathcal{N}_m} x_j$ and send $q_z^2$ to the active client $\mathcal{P}_{c(i)}$. $\mathcal{P}_{c(i)}$ then sum all received information to get $z_i^2 := \sum_{m \in \mathcal{N}_i^{cp}} \sum_{j \in \mathcal{N}_m} x_j$. By minis $i$'s information, $\mathcal{P}_{c(i)}$ gets $f_i^2 := z_i^2 - |\mathcal{N}_i^{cp}| x_i$, which is the sum of features of nodes in the third layer of the rooted tree, where $|\cdot|$ represents the size of a set.

Given $f_i^1$ and $f_i^2$, we design a decoder $\phi$ to extract features of nodes in the second layer and third layer of the rooted tree simultaneously. Specifically, the decoder is designed to achieve the following capability:

$$\phi(\text{CONCAT}(f_i^1, f_i^2)) = \{\text{CONCAT}(\hat{x}_m, \hat{H}_{\mathcal{N}_m \setminus \{i\}}) | m \in \mathcal{N}_i^{cp}\}, \tag{3}$$

where $\hat{H}_{\mathcal{N}_m \setminus \{i\}} = \sum_{j \in \mathcal{N}_m \setminus \{i\}} \hat{x}_j$ is the neighbour information of $m$ except for $i$. According to the dimension of features, we can split the outputs into two sets $G_i = \{\hat{x}_m | m \in \mathcal{N}_i^{cp}\}$ and $H_i = \{\hat{H}_{\mathcal{N}_m \setminus \{i\}} | m \in \mathcal{N}_i^{cp}\}$ corresponding to features of nodes in the second and third layer of the rooted tree. The rooted tree can be easily constructed by connecting nodes in $G_i$ to the rooted node and nodes in $H_i$ to the corresponding node in $G_i$.

**Generalizing to K-hop neighborhood reconstruction**

Following the derivation for the 1-hop and 2-hop cases, one can similarly generalize the method to reconstruct $K$-hop neighborhood. For every node $i \in \mathcal{V}$, denote $\mathcal{R}_i^k, k \in [K]$ as the set of neighbor nodes in the $k+1$-th layer of the rooted tree for the missing neighborhood. To construct the rooted tree, we first get the sum of features of nodes in each layer $f_i^k = \sum_{m \in \mathcal{R}_i^k} x_m, k \in [K]$, and then input them to the decoder to reconstruct the entire neighborhood information (Figure 2). Meanwhile, denote $\mathcal{N}_i^k, k \in [K]$ as the set of neighbor nodes in the $k+1$-th layer of the rooted tree for the entire neighborhood, $\mathcal{R}_i^k \subset \mathcal{N}_i^k$. We further denote $\mu_i^k = \sum_{m \in \mathcal{N}_i^k} x_m, k \in [K]$.

Protocol 1 in the Appendix F depicts the process of getting $f_i^k$. The core idea is that the $k$-hop neighbor information of any node $i$ is contained in the $k-1$-hop neighbor information of $i$'s neighbor nodes. To that end, we only require passive participants send information to the active participant. Specifically, for every node $i \in \mathcal{V}$, denote $\mathcal{C}_i = \{\mathcal{P}_p | \exists m \in \mathcal{N}_i^{cp}, \text{s.t}, p = c(m)\}$ as the set of passive participants for node $i$. We first get $f_i^1$ (line 3-8) and $f_i^2$ (line 13-20) following the process described above. Meanwhile, We also get $\mu_i^1, \eta_i^1 := \mu_i^1 - f_i^1$ (line 9-10) and $\mu_i^2, \eta_i^2 := \mu_i^2 - f_i^2$ (line 19-20), where $\eta_i$ denotes neighbor information without loss for node $i$. To construct $f_i^3$, we have passive client $\mathcal{P}_p \in \mathcal{C}_i$ sends $q_p^3 := \sum_{m \in \mathcal{N}_i^{cp}} \mathbb{1}_p[c(m)] \mu_m^2$ to the active client. After summing all received information, it is then required to minus $(|\mathcal{N}_i^{cp}| - 1) \cdot f_i^1$, wherein $f_i^1$ is missed 1-hop neighbor information that already exists in the second layer of the rooted tree. Meanwhile, as each $q_p^3$ contains $i$'s information without loss, it is required minus $|\mathcal{N}_i^{cp}| \cdot \eta_i^1$ and finally get $f_i^3 = \sum_{\mathcal{P}_p \in \mathcal{C}_i} q_p^3 - (|\mathcal{N}_i^{cp}| - 1) \cdot f_i^1 - |\mathcal{N}_i^{cp}| \cdot \eta_i^1$ (line 22). Similarly, $\mu_i^3 = \sum_{\mathcal{P}_p \in \mathcal{C}_i \cup \{c(i)\}} q_p^3 - (|\mathcal{N}_i^{cp}| + |\mathcal{N}_i^p| - 1) \cdot \mu_i^1$ (line 23), and $\eta_i^3 = \mu_i^3 - f_i^3$ (line 24). Repeating the above procedure, we can get $f_i^k$ for any $k > 3$.

The decoding process is challenging, as we need to decode feature information of nodes in the rooted tree and reconstruct the edges between them. To achieve such a goal, we propose an algorithm to reconstruct the neighborhood with the help of multiple decoders $\phi_k, k \in [K - 1]$, where $\phi_k$ can extract features of $k$-hop neighborhood information. In more detail, $\phi_k$ is designed to achieve the following capability:

$$\phi_k \left( \text{CONCAT} \left( f_i^k, ..., f_i^K \right) \right)$$
$$= \{ \text{CONCAT}(\hat{x}_{l_k}, \sum_{l_{k+1} \in \mathcal{N}_{l_k} \backslash \{l_{k-1}\}} \hat{x}_{l_{k+1}}, ..., \sum_{l_{k+1} \in \mathcal{N}_{l_k} \backslash \{l_{k-1}\}} ... \sum_{l_K \in \mathcal{N}_{l_{K-1}} \backslash \{l_{K-2}\}} \hat{x}_{l_K}) \mid l_k \in \mathcal{N}_{k-1}^{cp} \},$$

(4)

where $l_k, k \in [K]$ corresponding to $k$-hop neighbor nodes.

Algorithm 2 in the Appendix F presents the pseudo-code of the construction process. We first concatenate $f_i^k, k \in [K]$ (line 4) and input it to $\phi_1$ (line 9). We split the outputs into two sets $G_i^1, H_i^1$, where $G_i^1 = \{\hat{x}_{l_1} | l_1 \in \mathcal{N}_i^{cp}\}$ and $H_i^1 = \{\text{CONCAT}(\sum_{l_2 \in \mathcal{N}_{l_1} \backslash \{i\}} \hat{x}_{l_2}, ..., \sum_{l_2 \in \mathcal{N}_{l_1} \backslash \{i\}} ... \sum_{l_K \in \mathcal{N}_{l_{K-1}} \backslash \{l_{K-2}\}} \hat{x}_{l_K}) | l_1 \in \mathcal{N}_i^{cp}\}$. The second layer of the rooted tree for $i$ is constructed by generating nodes with features corresponding to elements in $G_i^1$. We then input each vector in $H_i^1$ to $\phi_2$ to decode features of nodes in the next layer (line 8-12). Specifically, for every $l_1 \in \mathcal{N}_i^{cp}$, we input the corresponding vector to $\phi_2$ and get two sets $G_i^2 = \{\hat{x}_{l_2} | l_2 \in \mathcal{N}_{l_1} \backslash \{i\}\}$ and $H_i^2 = \{\text{CONCAT}(\sum_{l_3 \in \mathcal{N}_{l_2} \backslash \{l_1\}} \hat{x}_{l_3}, ..., \sum_{l_3 \in \mathcal{N}_{l_3} \backslash \{l_1\}} ... \sum_{l_K \in \mathcal{N}_{l_{K-1}} \backslash \{l_{K-2}\}} \hat{x}_{l_K}) | l_2 \in \mathcal{N}_{l_1} \backslash \{i\}\}$. The third layer is constructed by generating nodes with features corresponding to elements in $G_i^2$ and connecting them to $l_1$. Repeat the above procedure, we can fully construct the first $K$ layers of the rooted tree. The last layer is constructed by directly generating nodes with features corresponding to elements in $H_i^K$ and connecting them to $l_{K-1}$.

### 4.3 DESIGN OF ENCODER-DECODER.

In the design of our encoder-decoder framework, we use the sum function as our encoder, which has the advantage of no need to learn. So the active client can learn an end-to-end decoder with the training dataset generated from her local graph. Although decoders $\phi_i, i \in [K]$ extract features of nodes in different layers separately, they are designed by the same principle, i.e., extract node' features and the corresponding neighbor information simultaneously. The main difference is that the

scope of the neighbor information is different, where $\phi_1$ extracts $K - 1$-hop neighbor information while $\phi_{K-1}$ extracts 1-hop neighbor information. To that end, it is only needed to train $\phi_1$ and apply it to extract features of nodes in all layers.

For ease of presentation, we show the training process of $\phi_1$ when $K = 2$. Specifically, we consider the following principle to reconstruct the 1-hop and 2-hop neighborhood information simultaneously. We have

$$\min_{\phi_1} \sum_{i \in \mathcal{V}} \mathcal{M}\left(Y_i, \phi_1(X_i)\right), \text{ s.t. } X_i = \sum_{h \in Y_i} h, \forall i \in \mathcal{V} \tag{5}$$

where $X_i = \text{CONCAT}(f_i^1, f_i^2)$, $Y_i = \left\{\text{CONCAT}\left(\hat{x}_m, \hat{H}_{\mathcal{N}_m \backslash \{i\}}\right) \mid m \in \mathcal{N}_i^{cp}\right\}$, and $\mathcal{M}(\cdot, \cdot)$ is the loss function that measures the loss to reconstruct a set $Y_i$. The task of learning $\phi_1$ is hard to characterize, and there are two fundamental challenges: First, as the distribution of node degrees is often long-tailed in real-world networks, the size of each set may vary differently. Second, a matching problem has to be solved to compare two equal-sized sets, which is costly if the set size is large. The first problem is easy to solve. As each node knows its neighbor nodes, the active client can sample at most $d$ neighbor nodes and ask passive clients to send the information of sample nodes only. For the second problem, we transform the decoding task into learning the probability distribution of the neighborhood information from a source distribution. Specifically, for node $i$, the neighborhood information is represented as an empirical realization of i.i.d sampling of $d_i$ elements from $\mathcal{P}_i$, where $\mathcal{P}_i \triangleq \frac{1}{d_i} \sum_{m \in \mathcal{N}_i} \text{CONCAT}(x_m, H_{\mathcal{N}_m \backslash \{i\}})$ and the source distribution $\mathcal{Q}_i \triangleq \sum_{m \in \mathcal{N}_i} \mathcal{P}_i^{(m)}$. Therefore, we adopt

$$\mathcal{M}\left(Y_i, \phi_1(X_i)\right) = \mathcal{W}_2^2\left(\mathcal{P}_i, \phi_1(X_i)\right), \tag{6}$$

where $\mathcal{W}_2$ is the 2-Wasserstein distance (Villani, 2009).

We make use of the architecture of a U-Net network (Ronneberger et al., 2015) to construct the decoder, which has the advantage of generating features of all neighbors in one forward pass. A theoretical analysis of the capability of the decoder is stated in Theorem 4.1 (See the proof in Appendix B, reproduced in Theorem 2.1 in (Lu & Lu, 2020)).

**Theorem 4.1.** *For any $\epsilon > 0$, if the support of the distribution $\mathcal{P}_i$ lies in a bounded space of $\mathbb{R}^d$, and $\mathcal{Q}_i$ is absolutely continuous with respect to the Lebesgue measure, there exists a feed-forward neural network $u(\cdot) : \mathbb{R}^d \to \mathbb{R}$ (and thus its gradient $\nabla u(\cdot) : \mathbb{R}^d \to \mathbb{R}^d$ ) with large enough width and depth (depending on $\epsilon$) such that $\mathcal{W}_2^2\left(\mathcal{P}_i, (\nabla u)_\# \mathcal{Q}_i\right) < \epsilon$.*

The decoder $\phi_1$ requires knowing the number of missing neighbors in advance, which is unavailable for the active client when applying it to extract features of $k$-hop ($k > 1$) neighbors. To that end, we use a predictor to predict the number of neighbors. Specifically, predictor $\phi_d$ takes $\hat{x}_m$ and $\hat{H}_{\mathcal{N}_m \backslash \{i\}}$ as inputs to predict the size of $|\mathcal{N}_m \backslash \{i\}|$. Denote $\mathcal{L}$ as the loss function. $\phi_d$ is joint training with $\phi_1$ by optimizing the following loss function:

$$\mathcal{M}\left(Y_i, \phi_1(X_i)\right) + \sum_{m \in \mathcal{N}_i^{cp}} \mathcal{L}(\phi_d(\hat{x}_m, \hat{H}_{\mathcal{N}_m \backslash \{i\}}), |\mathcal{N}_m \backslash \{i\}|). \tag{7}$$

In practice, we adopt the empirical Wasserstein distance that can provably approximate the population one. For node $i$, for every forward pass, the model will get $|Y_i|$ outputs denoted as $\hat{\xi}_1, \ldots, \hat{\xi}_{|Y_i|}$. Inspired by the work of (Tang et al., 2022), which propose an auto-encoder framework for graph data, we adopt the surrogated loss defined as follow:

$$\min_{\pi \in \Pi} \sum_{j=1}^{|Y_i|} \left\| \xi_i^j - \hat{\xi}_i^{\pi(j)} \right\|^2, \text{ s.t. } \pi \text{ is a bijective mapping: } [|Y_i|] \to [|Y_i|]. \tag{8}$$

Meanwhile, the original node features could be high-dimensional, and reconstructing them directly may introduce a lot of noise. Instead, we may first map node features into a latent space. In more detail, all clients could collectively learn a model and map features to a low dimension with linear layers in the model.

## 5 EXPERIMENTS

In this section, we conduct extensive experiments to evaluate Fed$^2$GNN focusing on the following research questions: **RQ1**: How does Fed$^2$GNN perform in comparison to state-of-the-art federated

Table 1: Summary of node classification accuracy results in percent (GCN, GraphSAGE).

| Model | Cora | | | MSAcademic | | | DBLP | | |
|---|---|---|---|---|---|---|---|---|---|
| **GCN** | M=5 | M=10 | M=15 | M=5 | M=1 | M=15 | M=5 | M=10 | M=15 |
| FedAvg | 86.02±0.55 | 81.39±1.55 | 77.88±1.96 | 93.03±0.15 | 90.78±0.64 | 87.94±0.61 | 84.29±0.32 | 82.02±0.31 | 79.57±0.77 |
| FedAvg-Full | 86.49±0.37 | 82.89±1.84 | 81.53±1.57 | 93.50±0.13 | 92.12±0.56 | 91.25±0.50 | 84.72±0.34 | 83.40±0.23 | 82.44±0.52 |
| **Fed$^2$GNN** | 86.15±0.51 | 83.30±1.76 | 82.20±1.15 | 93.59±0.14 | 92.12±0.43 | 91.81±0.59 | 84.90±0.18 | 84.02±0.08 | 83.42±0.19 |
| Central | | 87.88 ±0.48 | | | 93.72 ±0.14 | | | 85.01 ±0.09 | |

| Model | Cora | | | MSAcademic | | | DBLP | | |
|---|---|---|---|---|---|---|---|---|---|
| **GraphSAGE** | M=5 | M=10 | M=15 | M=5 | M=10 | M=15 | M=5 | M=10 | M=15 |
| FedAvg | 84.64±0.49 | 81.37±0.85 | 77.62±1.08 | 93.29±0.16 | 91.17±0.69 | 89.06±0.80 | 83.82±0.27 | 82.43±0.11 | 81.54±0.15 |
| FedAvg-Full | 85.19±0.71 | 83.41±0.79 | 80.84±0.60 | 93.40±0.40 | 91.99±0.74 | 90.53±1.03 | 84.25±0.23 | 83.74±0.18 | 83.00±0.16 |
| FedSage+ | 84.66±0.75 | 83.54±0.70 | 82.39±0.59 | 92.59±0.22 | 91.32±0.05 | 89.97±0.16 | 79.96±0.59 | 80.79±0.69 | 81.18±0.79 |
| **Fed$^2$GNN** | 86.07±0.99 | 84.22±0.76 | 80.46±0.69 | 93.14±0.55 | 91.04±1.37 | 90.01±1.24 | 84.28±0.18 | 83.36±0.26 | 83.07±0.13 |
| Central | | 87.15 ±0.60 | | | 94.76 ±0.19 | | | 84.51 ±0.23 | |

Table 2: Summary of node classification accuracy results in percent (5-layer deep GNN).

| Model | Cora | | | MSAcademic | | | DBLP | | |
|---|---|---|---|---|---|---|---|---|---|
| **DAGNN** | M=5 | M=10 | M=15 | M=5 | M=10 | M=15 | M=5 | M=10 | M=15 |
| FedAvg | 87.62±0.54 | 84.04±0.73 | 81.42±0.90 | 93.71±0.26 | 92.63±0.26 | 90.57±0.81 | 84.49±0.07 | 83.26±0.07 | 82.38±0.15 |
| FedAvg-Full | 87.65±0.45 | 86.79±0.55 | 85.20±0.97 | 94.15±0.26 | 93.40±0.42 | 92.57±0.57 | 84.74±0.09 | 84.34±0.15 | 84.41±0.10 |
| FedSage+ | 83.87±1.14 | 83.06±0.75 | 83.74±1.14 | 92.31±0.43 | 91.24±0.24 | 91.00±0.06 | 80.44±0.26 | 80.74±0.50 | 80.80±0.28 |
| **Fed$^2$GNN** | 87.41±0.20 | 86.78±0.77 | 86.72±1.02 | 94.01±0.01 | 93.65±0.47 | 92.89±0.36 | 84.56±0.20 | 84.16±0.21 | 83.91±0.09 |
| Central | | 88.15 ±0.54 | | | 94.56 ±0.14 | | | 85.04 ±0.35 | |

graph learning baselines for training 2-layer GNNs? **RQ2**: How does Fed$^2$GNN perform on deep GNNs compared to baselines? **RQ3**: What are the impacts of the max node degree $d$ to Fed$^2$GNN?

## 5.1 EXPERIMENTAL SETTING

We conduct experiments on three citation network datasets, Cora (Sen et al., 2008), MSAcademic (Shchur et al., 2018), and DBLP (Fu et al., 2020). We randomly split all datasets with 40% training set, 30% validation set, and 30% testing set. To synthesize the distributed subgraph system, we generate hierarchical graph clusters on each dataset with the Louvain algorithm (Blondel et al., 2008) following the work of FedSage+ (Zhang et al., 2021). We consider three scenarios of $M = 5, 10, 15$. The statistics of datasets are presented in Appendix C.

We compare Fed$^2$GNN with four baselines to demonstrate its effectiveness. All experiments were repeated ten times with different random seeds.

- **Central learning**: models are trained in a centralized manner with a global graph dataset.
- **FedAvg**: models are trained by utilizing the FedAvg algorithm on distributed subgraph data.
- **FedAvg-Full**: models are trained by utilizing the FedAvg algorithm, but the node representations have no neighborhood information loss. This baseline is in line with FedGCN (Yao & Joe-Wong, 2022)'s goal in terms of accuracy and represents the target of our framework.
- **FedSage+** (Zhang et al., 2021): a federated graph learning framework with the idea of generating cross-subgraph neighbor nodes.

## 5.2 PERFORMANCE FOR 2-LAYER GNNS.(RQ1)

We conduct experiments on two GNN models: GCN (Kipf & Welling, 2017) and GraphSAGE (Hamilton et al., 2017b), and compare the performance of Fed$^2$GNN with all baselines. The results are depicted in Table 1. It shows that Fed$^2$GNN has the best performance over FedAvg and is better than FedSage+ in most cases. Meanwhile, Fed$^2$GNN has a comparable performance with FedAvg-full, corroborating the effectiveness of our method for reconstructing neighborhood information.

## 5.3 PERFORMANCE FOR DEEP GNNS (RQ2)

We conduct experiments to evaluate the performance of our framework for deep GNNs. We adopt the model DAGNN proposed in (Liu et al., 2020), which has a better performance for node representation learning than GCN (Kipf & Welling, 2017) on many dataset. Table 2 presents the results

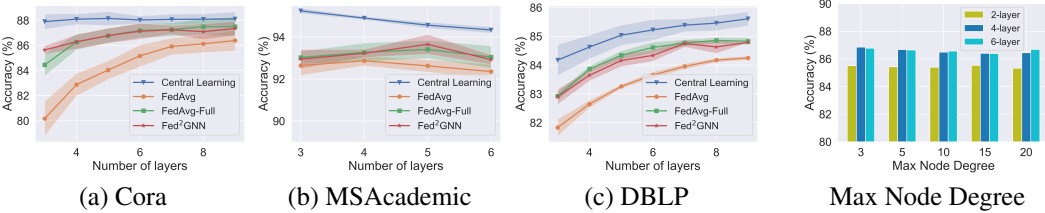

(a) Cora      (b) MSAcademic      (c) DBLP      Max Node Degree

Figure 3: Results of Fed$^2$GNN for training deep GNNs with different number of layers.

Figure 4: Accuracy v.s. Max Node Degree

when the number of layers is 5. It shows that Fed$^2$GNN still has a good performance even for deep GNNs, demonstrating the effectiveness of our methods for reconstructing neighborhood information. Meanwhile, another important observation emerging from the results is that learning for DAGNN significantly bridges the gap between central learning and federated learning on the Cora dataset. Specifically, as can be observed in Table 1, the gaps between central learning and Fed$^2$GNN is 4.58% when $M = 10$ and 5.68% when $M = 15$ for GCN model, while the gaps are only 1.37% and 1.43% for deep GNNs. Such an observation assay the benefits brought by DAGNN in the federated setting.

We further conduct experiments for DAGNN with different depths. The results are illustrated in Figure 3. It shows that node representation learning on Cora and DBLP datasets benefits from training the DAGNN model. Meanwhile, applying DAGNN on the MSAcademic dataset degrades the performance in central learning. Both of the observations are aligned with the results in (Liu et al., 2020). However, federated graph learning on the MSAcademic dataset could benefit from applying deeper DAGNN, probably because deep GNNs have a better generalization ability and compensate for the federated learning. Moreover, Fed$^2$GNN still performs better than FedAvg and has comparable performance with FedAvg-full, showing that our method is effective for federated learning of deep GNNs.

## 5.4 IN-DEPTH ANALYSIS FOR FED$^2$GNN(RQ3)

In Figure 4, we present results on studying the impact of max node degree $d$ for Fed$^2$GNN. The size of $d$ not only affects the efficiency of reconstructing neighborhood information but also affects of performance of downstream tasks. We experiment on models with a different number of layers with the Core dataset and try $q$ varies from 3 to 20. It shows that the performance of Fed$^2$GNN is robust to the max node degree, with large $d$ only leading to slightly better performance them small $d$.

Finally, to further understand the effectiveness of our proposed framework for federated graph learning, we perform convergence analysis in Appendix E. The results show that the method has a similar converge rate as FedAvg-full, indicating that we can indeed recover the missing neighborhood. Meanwhile, we conduct experiments on training the decoder locally or collectively; the results show that the decoder has a good generalization ability.

## 6 CONCLUSION

In this work, we address the limitations of existing works of federated learning on graph-structured data and propose a new framework that can better tackle the issue of missing cross-client neighborhood information during training. Our framework utilizes the idea of reconstructing neighborhood information that considers both structured-based and feature-based information. Such an advantage allows us to train deep GNNs in the federated setting, leading to better performance for node representation learning. Extensive experiments have been conducted to verify the effectiveness of our framework, which is consistent with our theoretical analysis. Though our framework manifests good performance, it confronts potential privacy concerns as other works in federated learning. Solving such a problem could be a promising direction in the future.

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

## A  FED$^2$GNN FOR GCN MODEL

The rooted tree recovers the structure-based information of one node. However, in practice, the last layer of the rooted tree is not fully splitted, which change the degrees of nodes in the last two layers. Specifically, the nodes in the last layer has a constant degree 2 (including self-loop) and the nodes in the second to last layer has a constant degree 3 (including self-loop), which affects the encoding process of models such as GCN (Kipf & Welling, 2017). Surprisingly, we can slightly change the encoding process and manipulate the edge weight to solve the problem.

Recall that the representation of a GCN model is formulated as $H_i^{(k)} = \sigma(\sum_{j \in \mathcal{N}_i \cup \{i\}} \frac{1}{\sqrt{(|\mathcal{N}_i|+1)\cdot(|\mathcal{N}_j|+1)}} H_j^{(k-1)} W^{(k)})$. In the 1-hop neighborhood reconstruction phase, instead of sending $\sum_{m \in \mathcal{N}_i^{cp}} \mathbb{1}_z[c(m)]x_m$ directly, the passive client $\mathcal{P}_z$ sends $\sum_{m \in \mathcal{N}_i^{cp}} \mathbb{1}_z[c(m)] \frac{x_m}{\sqrt{|\mathcal{N}_m|+1}}$ to the active client, and $\sum_{m \in \mathcal{N}_i^{cp}} \mathbb{1}_z[c(m)] \sum_{j \in \mathcal{N}_m} \frac{x_j}{\sqrt{(|\mathcal{N}_j|+1)}}$ in the 2-hop neighborhood reconstruction phase. With a decoder, we get two set $\mathcal{G}_i = \{\frac{\hat{x}_m}{\sqrt{|\mathcal{N}_m|+1}} | m \in \mathcal{N}_i^{cp}\}$ and $\mathcal{H}_i = \{\sum_{j \in \mathcal{N}_m \setminus \{i\}} \frac{\hat{x}_j}{\sqrt{|\mathcal{N}_j|+1}} | m \in \mathcal{N}_i^{cp}\}$. We represent the 1-hop neighbors as $\frac{\sqrt{3} \cdot \hat{x}_m}{\sqrt{|\mathcal{N}_m|+1}}$ and 2-hop neighbors as $\sum_{j \in \mathcal{N}_m \setminus \{i\}} \frac{\sqrt{2} \cdot \hat{x}_j}{\sqrt{(|\mathcal{N}_j|+1)}}, \forall m \in \mathcal{N}_i^{cp}$.

We further manipulate the edge weight between node $i$ and the generated neighbor nodes including 1-hop neighbors $m \in \mathcal{N}_i^{cp}$ and the corresponding 2-hop neighbor $j_m$ connecting to $m$. For pair of nodes $A, B$, we denote the message passing from $A$ to $B$ as $A \to B$. In undirectd graphs, message passing between two nodes bidirectionally, hence we present the weights of $A \to B$ and $B \to A$ separately. We have the following results:

| Direction | Weight | Direction | Weight |
|-----------|--------|-----------|--------|
| $i \to m$ | $\frac{\sqrt{3}}{(|\mathcal{N}_i|+1)\cdot(\sqrt{|\mathcal{N}_m|+1})}$ | $m \to i$ | $\frac{1}{\sqrt{3}\sqrt{|\mathcal{N}_i|+1}}$ |
| $i \to i$ | $\frac{1}{|\mathcal{N}_m|+1}$ | $j_m \to m$ | $\frac{\sqrt{3}}{\sqrt{2}(|\mathcal{N}_m|+1)}$ |

Note that the message passing from $m \to j_m$ does not affect the encoding process for note $i$; we omit it here.

*Proof.* For easy of presentation, for every node $v \in \mathcal{V}$, we denote $D_v$ as the degree of $v$ including self loop. For a two layer GCN model, we have:

$$
\begin{aligned}
\hat{y}_i &= \sigma_1 \big( \sum_{m \in \mathcal{N}_i \cup \{i\}} \frac{1}{\sqrt{D_i \cdot D_m}} \sigma_2 \big( \sum_{j \in \mathcal{N}_m \cup \{m\}} \frac{1}{\sqrt{D_m \cdot D_j}} x_j W^{(1)} \big) W^{(2)} \big) \\
&= \sigma_1 \big( \big( \frac{1}{D_i} \sigma_2 \big( \big( \frac{1}{D_i} x_i + \sum_{j \in \mathcal{N}_i} \frac{1}{\sqrt{D_i \cdot D_m}} x_j \big) W^{(1)} \big) \\
&\quad + \sum_{m \in \mathcal{N}_i} \frac{1}{\sqrt{D_i \cdot D_m}} \sigma_2 \big( \big( \frac{1}{D_m} x_m + \sum_{j \in \mathcal{N}_m} \frac{1}{\sqrt{D_m \cdot D_j}} x_j \big) W^{(1)} \big) \big) W^{(2)} \big)
\end{aligned}
\tag{9}
$$

In our methods, we manipulate the edge weight and features of neighbors, such that encoding result is the same as Equation 9.

$$
\begin{aligned}
\hat{y}_i &= \sigma_1 \big( \big( \frac{1}{D_i} \sigma_2 \big( \big( \frac{1}{D_i} x_i + \sum_{j \in \mathcal{N}_i} \frac{1}{\sqrt{3 D_i}} \cdot \frac{\sqrt{3} x_j}{\sqrt{D_m}} \big) W^{(1)} \big) \\
&\quad + \sum_{m \in \mathcal{N}_i} \frac{1}{\sqrt{3 D_i}} \sigma_2 \big( \big( \frac{1}{D_m} \frac{\sqrt{3} x_m}{\sqrt{D_m}} + \frac{\sqrt{3} x_i}{D_i \cdot \sqrt{D_m}} \\
&\quad + \frac{\sqrt{3}}{\sqrt{2} D_m} \sum_{j \in \mathcal{N}_m \setminus \{i\}} \frac{\sqrt{2} x_j}{\sqrt{D_j}} \big) W^{(1)} \big) \big) W^{(2)} \big)
\end{aligned}
\tag{10}
$$

Note that $\sigma_2$ is the ReLU function, rearranging the above formula, we can get Equation 9. $\square$

## B    PROOF OF THEOREM 4.1

Theorem 4.1 is reproduced from Theorem 2.1 in Lu & Lu (2020), which is stated as below:

**Theorem B.1.** *(Theorem 2.1 in Lu & Lu (2020)). Let $\mathcal{P}$ and $\mathcal{Q}$ be the target and the source distributions respectively, both defined on $\mathbb{R}^d$. Assume that $\mathcal{Q}$ is absolutely continuous with respect to the Lebesgue measure and $\mathbf{E}_{X \sim \mathcal{P}} |X|^3 < \infty$, it holds that for any given approximation error $\varepsilon$, there exists a positive integer $n = O(\frac{1}{\varepsilon^d})$, and a fully connected and feed-forward deep neural network $u(\cdot)$ of depth $L = \lceil \log_2 n \rceil$ and width $N = 2^L = 2^{\lceil \log_2 n \rceil}$, with $d$ inputs and a single output and with ReLU activation such that $\mathcal{W}_1 \left( (\nabla u)_\# \mathcal{Q}, \mathcal{P} \right) \leq \varepsilon$.*

To prove Theorem 4.1, we only need to verify the conditions presented in the above theorem. We need to show that $\mathcal{P} = \mathcal{P}_i$ has a bound $\mathbf{E}_{X \sim \mathcal{P}} |X|^3 < \infty$ and $\mathcal{Q} = \mathcal{Q}_i$ is absolutely continuous with respect to the Lebesgue measure. Furthermore, we also need to show the connection between $\mathcal{W}_1(\cdot, \cdot)$ and $\mathcal{W}_2(\cdot, \cdot)$.

*Proof.* $\mathcal{P} = \mathcal{P}_i$ has a bounded 3-order moment because the support of $\mathcal{P}$ is in a bound space of $\mathcal{R}^d$

$\mathcal{Q} = \mathcal{Q}_i$ is absolutely continuous with respect to the Lebesgue measure because $\mathcal{P}_i$ is absolutely continuous with respect to the Lebesgue measure.

Last, we show the connection between $\mathcal{W}_1(\cdot, \cdot)$ and $\mathcal{W}_2(\cdot, \cdot)$. Note that the support $\mathcal{P} = \mathcal{P}_i$ is bounded, i.e., $\forall \delta \in supp(\mathcal{P})$, there exist $C \in \infty$, s.t. $||\delta|| < C$. According to (Lu & Lu, 2020), $\tilde{\mathcal{Q}} = (\nabla u)_\# \mathcal{Q}$ also has bounded support. Wlog, we have $\forall \delta \in supp(\tilde{\mathcal{Q}}), ||\delta|| < C$. Then, we show that

$$\mathcal{W}_2^2(\mathcal{P}, \tilde{\mathcal{Q}}) = \inf_{\gamma \in \Gamma(\mathcal{P}, \tilde{\mathcal{Q}})} \left[ \int_{\mathcal{Z} \times \mathcal{Z}'} \|Z - Z'\|_2^2 \, d\gamma(Z, Z') \right]$$

$$\leq 2C \inf_{\gamma \in \Gamma(\mathcal{P}, \tilde{\mathcal{Q}})} \left[ \int_{\mathcal{Z}' \mathcal{Z}'} \|Z - Z'\|_2 \, d\gamma(Z, Z') \right]$$

$$\leq 2C\sqrt{d_x} \inf_{\gamma \in \Gamma(\mathcal{P}, \tilde{\mathcal{Q}})} \left[ \int_{\mathcal{Z} \times \mathcal{Z}'} \|Z - Z'\|_1 \, d\gamma(Z, Z') \right]$$

$$= 2C\sqrt{d_x} \mathcal{W}_1(\mathcal{P}, \tilde{\mathcal{Q}})$$

$$< 2C\sqrt{d_x}\epsilon.$$

As $C$ and $\sqrt{d_x}$ are constant, we have $\mathcal{W}_2^2(\mathcal{P}, \tilde{\mathcal{Q}}) = O(\varepsilon)$. The first inequality is because $\|Z - Z'\|_2 \leq \|Z\|_2 + \|Z'\|_2 = 2C$. The second inequality is because

$$\sqrt{d_x} \inf_{\gamma \in \Gamma(P, Q)} \left[ \int_{\mathcal{Z} \times \mathcal{Z}'} \|Z - Z'\|_1 \, d\gamma(Z, Z') \right]$$

$$= \lim_{i \to \infty} \left[ \int_{\mathcal{Z} \times \mathcal{Z}'} \sqrt{d_x} \|Z - Z'\|_1 \, d\gamma_i(Z, Z') \right]$$

(There exists a sequence of measures $\{\gamma_i\}_{i=1}^{\infty}$ achieving the infimum)

$$\geq \lim_{i \to \infty} \left[ \int_{\mathcal{Z} \times \mathcal{Z}'} \|Z - Z'\|_2 \, d\gamma_i(Z, Z') \right]$$

$$\geq \inf_{\gamma \in \Gamma(P, Q)} \left[ \int_{\mathcal{Z} \times \mathcal{Z}'} \|Z - Z'\|_2 \, d\gamma(Z, Z') \right].$$

$\square$

## C  DATASET STATISTICS

In this section, we present details of the datasets used in our experiments. We conduct experiments on three citation network datasets, Cora (Sen et al., 2008), MSAcademic (Shchur et al., 2018), and DBLP (Fu et al., 2020). We summarize the statistics of the datasets in Table 3. Meanwhile, we also summarize the statistics of subgraphs obtained by the Louvain algorithm (Blondel et al., 2008) in Table 4.

Table 3: Statistics of the datasets, where $|\mathcal{V}|$ represents the number of nodes, $|\mathcal{E}|$ represents the number of edges, $|\mathcal{X}|$ represents the number of features and Classes is the number of label classes.

| Dataset | $|\mathcal{V}|$ | $|\mathcal{E}|$ | $|\mathcal{X}|$ | Classes |
|---------|------|------|------|---------|
| Cora | 2,708 | 5,429 | 1,433 | 7 |
| MSAcademic | 18,333 | 81,894 | 6,805 | 15 |
| DBLP | 17,716 | 52,867 | 1,639 | 4 |

Table 4: Statistics of subgraphs, where $|\mathcal{V}|$ represents the average number of nodes, $|\mathcal{E}|$ represents the average number of edges and $\Delta|\mathcal{E}|$ is the total number of the cross-client edges.

| Dataset | Cora | | | MSAcademic | | | DBLP | | |
|---------|------|------|------|------------|------|------|------|------|------|
| | M=5 | M=10 | M=15 | M=5 | M=10 | M=15 | M=5 | M=10 | M=15 |
| $|\mathcal{V}|$ | 542 | 271 | 181 | 3667 | 1833 | 1222 | 3543 | 1772 | 1181 |
| $|\mathcal{E}|$ | 946 | 443 | 254 | 13933 | 6099 | 3409 | 8898 | 4358 | 2826 |
| $\Delta|\mathcal{E}|$ | 697 | 1002 | 1613 | 12230 | 20906 | 30765 | 8379 | 9287 | 10471 |

## D  HYPER-PARAMETER SETTING

In Table 5, we present the descriptions of hyper-parameters and the range of them used in our experiments. The details of hyper-parameters settings used in experiments are shown in Table 6.

Table 5: Hyper-parameter range of GNN and Decoder

| Hyper-parameter | Description | Range |
|---|---|---|
| Learning rate | The learning rate for Adam optimization | [0.1, 0.01, 0.001] |
| Dropout rate | The Dropout rate | [0.0, 0.25, 0.50, 0.75] |
| L2 regularization | The L2 regularization | $[1 \cdot 10^{-3}, 1 \cdot 10^{-4}]$ |
| Hidden dimension | The hidden layer dimensions of GNN | [16, 32] |
| $M$ | Number of clients in federated learning | [5, 10, 15] |
| Epoch | Training epochs, also federated learning rounds | 200 |
| Delta | Louvain split parameter | 40 |

| Hyper-parameter | Description | Range |
|---|---|---|
| $d$ | Max node degree | [3, 5, 10, 15, 20] |
| Pool size | The pool size of Decoder | [2, 3, 4] |
| Blocks | Blocks number of Separator | [4, 5] |
| Separator lr | Learning rate of Separator training | $1 \cdot 10^{-3}$ |
| Predictor lr | Learning rate of Predictor training | $1 \cdot 10^{-2}$ |
| Decoder epoch | training epochs of Decoder | [20, 50, 100, 200] |

Table 6: Hyper-parameter setting

| Hyper-parameter | GCN | | | GraphSAGE | | |
|---|---|---|---|---|---|---|
| | Cora | MSAcdemic | DBLP | Cora | MSAcdemic | DBLP |
| Learning rate | 0.01 | 0.1 | 0.01 | 0.01 | 0.01 | 0.01 |
| Dropout rate | 0.5 | 0.0 | 0.75 | 0.5 | 0.0 | 0.75 |
| L2 regularization | $1 \cdot 10^{-3}$ | $1 \cdot 10^{-3}$ | $1 \cdot 10^{-4}$ | $1 \cdot 10^{-3}$ | $1 \cdot 10^{-3}$ | $1 \cdot 10^{-4}$ |
| Hidden dimension | 16 | 16 | 32 | 16 | 16 | 32 |
| Epoch | 200 | 200 | 200 | 200 | 200 | 200 |
| Delta | 40 | 40 | 40 | 40 | 40 | 40 |
| $d$ | 10 | 10 | 10 | 10 | 10 | 10 |
| Pool size | 3 | 3 | 2 | 3 | 3 | 3 |
| Blocks | 4 | 4 | 5 | 5 | 5 | 5 |
| Separator lr | $1 \cdot 10^{-3}$ | $1 \cdot 10^{-3}$ | $1 \cdot 10^{-3}$ | $1 \cdot 10^{-3}$ | $1 \cdot 10^{-3}$ | $1 \cdot 10^{-3}$ |
| Predictor lr | $1 \cdot 10^{-2}$ | $1 \cdot 10^{-2}$ | $1 \cdot 10^{-2}$ | $1 \cdot 10^{-2}$ | $1 \cdot 10^{-2}$ | $1 \cdot 10^{-2}$ |
| Decoder epoch | 200 | 200 | 200 | 50 | 20 | 100 |

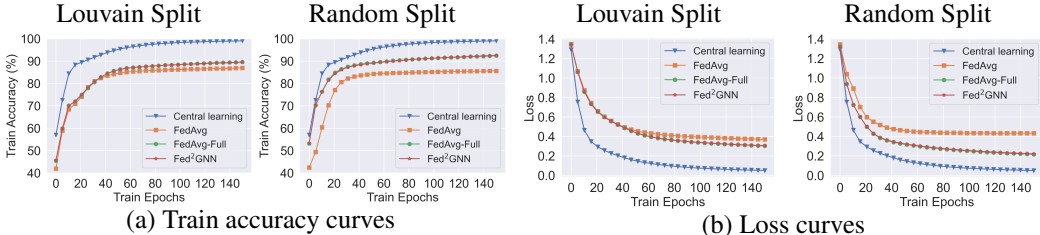

(a) Train accuracy curves          (b) Loss curves

Figure 5: Average accuracy curves and loss curves on DBLP, $M = 10$

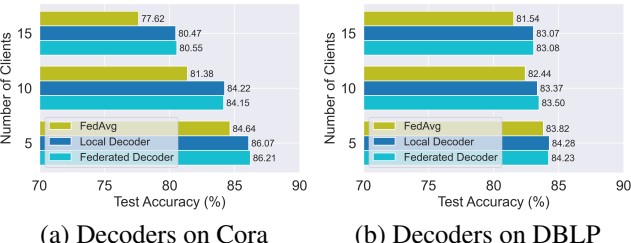

(a) Decoders on Cora        (b) Decoders on DBLP

Figure 6: Decoder training methods study

## E ADDITIONAL EXPERIMENTS

### E.1 CONVERGENCE STUDY

In this subsection, we repeat five times experimental runs and show the average (a) train accuracy curves and (b) loss curves in Figure 5 to demonstrate the convergence of our model. In addition to the Louvain split method applied in the previous work (Zhang et al., 2021), we also apply the random split method, which randomly selects nodes to form subgraphs and maintains only the edges between the selected nodes in the subgraph.

The results show that Fed²GNN model achieves almost the same convergence as the FedAvg-Full baseline. Since nodes are randomly selected to form subgraphs, there will be more cross-client edges in the subgraphs under the random split method. Therefore, our experimental results will be more evident under the random split method.

### E.2 DECODER TRAINING METHODS STUDY

In this subsection, we study the training methods of the decoder by comparing the test accuracy obtained under different decoder training methods. We tried locally trained decoders and federated trained decoders and set baselines FedAvg for comparison. We conduct experiments on Cora and DBLP datasets using the hyper-parameters setting in Table6. As shown in Figure 6, the federated trained decoder only slightly better that the locally trained decoder, showing that the decoder has a good generalization ability to decode information from unseen data.

### E.3 COMMUNICATION COST

Communication cost is widely recognized as a major bottleneck for federated learning. Particularly in federated graph learning, additional communication across clients may be required to recover missing neighborhood information. Recall that each client recursively aggregates information from its neighbor nodes during the information transmitting process in our methods; the communication is $\mathcal{O}(ndK)$, where $n$ is the number of nodes, $d$ is the average degree of all nodes, and $K$ is the number of layers of a GNN model.

We conduct experiments to evaluate the communication cost on real-world datasets with the baselines of FedGCN (Yao & Joe-Wong, 2022) and FedGraph (Chen et al., 2022), which have the communication cost of $\mathcal{O}(nd^k)$ and $\mathcal{O}(ndT)$, where $T >> k$ is the number of iterations. We ignore the influence of feature dimension since all methods can reduce the feature dimension in advance. The results are presented in Figure 7.

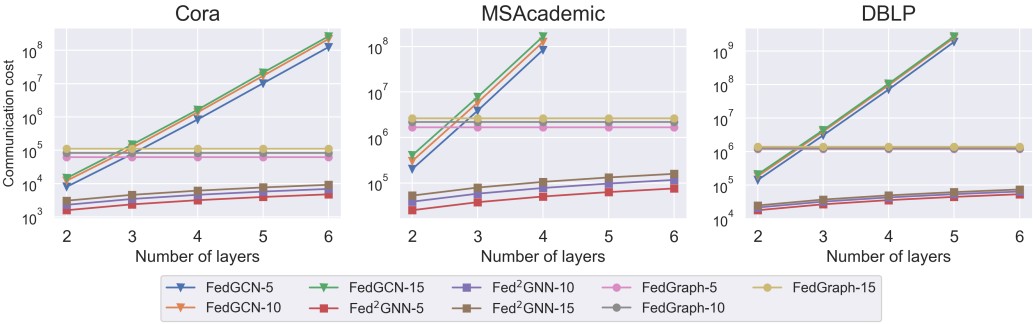

Figure 7: Communication cost of Fed²GNN in comparison with FedGCN (Yao & Joe-Wong, 2022) and FedGraph (Chen et al., 2022) for training models with different number of layers and different number of clients.

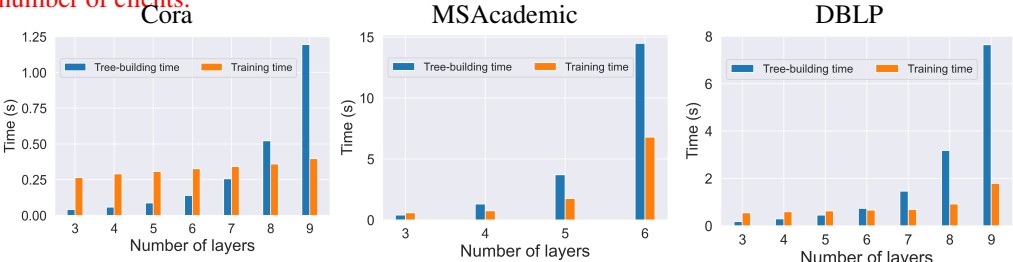

Figure 8: Average tree-building time and training time of all clients.

We vary the number of GNN layers and the number of clients. It shows that our methods have the lowest cost in all settings. Meanwhile, partitioning more clients requires more communication costs since more nodes have missing neighborhood information.

### E.4 EFFICIENCY OF CONSTRUCTING ROOTED TREES

Constructing the rooted tree for a node is simple. Algorithm 2 presents the pseudo-code of the process for one tree. It mainly consists of a sequence of predicting processes of the decoder $\phi$. For each vector $h \in H^k$, the decoder takes it as input and outputs $G$ and $H$ (line 9 in Algorithm 2). The $K+1$-th layer of the tree is constructed by generating nodes with features corresponding to elements in $G$ and connecting them to $g$ corresponding to $h$.

In practice, the vectors in the set $H^k$ can be fed to the decoder simultaneously. Meanwhile, we can even input all $H^1$s (of different root nodes) into the decoder to construct the rooted trees simultaneously. To that end, building rooted trees of $K$ layers for all nodes requires the decoder to predict only $K-1$ times.

We conduct experiments to evaluate the efficiency of the tree-building process and report the average tree-building time and training time for all clients. All experiments are implemented on a server equipped with an A100 GPU. The results are presented in Figure 8. It shows that the tree-building time increases as the number of layers grows. Indeed, as the number of layers increases, the number of generated nodes increases exponentially, hence requiring more time. However, it is still efficient to build rooted trees for all nodes, which only requires a few seconds.

## F ADDITIONAL PROTOCOL AND ALGORITHM

---

**Protocol 1:** Neighbor information transmission

---

1    **Input:** $\mathcal{P}_p$ has data $\mathcal{G}_p = (\mathcal{V}_p, \mathcal{E}_p, \mathcal{X}_p), p \in [M], \mathcal{X}_p = \{x_i, \forall i \in \mathcal{V}_p\}$; depth $K$; neighborhood function $\mathcal{N} : i \to 2^{\mathcal{V}}$; the set of passive participants $\mathcal{C}_i = \{\mathcal{P}_p \,|\, \exists m \in \mathcal{N}_i^{cp}, \text{s.t}, p = c(m)\}$ for each $i \in \cup_{p=1}^{M} \mathcal{V}_p$

2    **Output:** Information of $k$-hop neighbors $f_i^k, k = [K], i \in \cup_{p=1}^{M} \mathcal{V}_p$.

3    **for** $i \in \cup_{p=1}^{M} \mathcal{V}_p$ **do**

4      $\mu_i^0, f_i^0 \leftarrow x_i$

5      **for** $\mathcal{P}_p \in \mathcal{C}_i$ **do**

6        $\mathcal{P}_p$ sends $q_p^1 \leftarrow \sum_{m \in \mathcal{N}_i^{cp}} \mathbb{1}_p[c(m)] \mu_m^0$ to $\mathcal{P}_{c(i)}$

7      **end**

8      $f_i^1 \leftarrow \sum_{\mathcal{P}_p \in \mathcal{C}_i} q_p^1$

9      $\mu_i^1 \leftarrow \sum_{\mathcal{P}_p \in \mathcal{C}_i \cup \{\mathcal{P}_{c(i)}\}} q_p^1$

10      $\eta_i^1 \leftarrow \mu_i^1 - f_i^1$

11    **end**

12    **for** $k = 2...K$ **do**

13      **for** $i \in \cup_{p=1}^{M} \mathcal{V}_p$ **do**

14        **for** $\mathcal{P}_p \in \mathcal{C}_i$ **do**

15          $\mathcal{P}_p$ sends $q_p^k \leftarrow \sum_{m \in \mathcal{N}_i^{cp}} \mathbb{1}_p[c(m)] \mu_m^{k-1}$ to $\mathcal{P}_{c(i)}$

16        **end**

17        **if** $k == 2$ **then**

18          $f_i^k \leftarrow \sum_{\mathcal{P}_p \in \mathcal{C}_i} q_p^k - |\mathcal{N}_i^{cp}| \cdot f_i^{k-2}$

19          $\mu_i^k \leftarrow \sum_{\mathcal{P}_p \in \mathcal{C}_i \cup \{\mathcal{P}_{c(i)}\}} q_p^k - (|\mathcal{N}_i^{cp}| + |\mathcal{N}_i^p|) \cdot \mu_i^{k-2}$

20          $\eta_i^k \leftarrow \mu_i^k - f_i^k$

21        **else**

22          $f_i^k \leftarrow \sum_{\mathcal{P}_p \in \mathcal{C}_i} q_p^k - (|\mathcal{N}_i^{cp}| - 1) \cdot f_i^{k-2} - |\mathcal{N}_i^{cp}| \cdot \eta_i^{k-2}$

23          $\mu_i^k \leftarrow \sum_{\mathcal{P}_p \in \mathcal{C}_i \cup \{c(i)\}} q_p^k - (|\mathcal{N}_i^{cp}| + |\mathcal{N}_i^p| - 1) \cdot \mu_i^{k-2}$

24          $\eta_i^k \leftarrow \mu_i^k - f_i^k$

25        **end**

26      **end**

27    **end**

---

---

**Algorithm 2:** Rooted tree construction

---

1  **Input:** Node $i$ and its neighbor information $f_i^1, f_i^2, ...., f_i^K$; decoders $\phi_k, k \in [K-1]$ trained in advance; An operator ZIP which iterate over two set in parallel and producing a set of tuples with an item from each one.

2  **Output:** The rooted tree for $i$.

3  $G^0 \leftarrow \{x_i\}$

4  $H^1 = \{\text{CONCAT}(f_i^1, f_i^2, ...., f_i^K)\}$

5  **for** $k = [K-1]$ **do**

6     $G^k \leftarrow \emptyset$

7     $H^{k+1} \leftarrow \emptyset$

8     **for** $(g^{k-1}, h^k) \in \text{ZIP}(G^{k-1}, H^k)$ **do**

9         $G, H \leftarrow \phi_k(h^k)$

10        $G^k \leftarrow G^k \cup G$

11       $H^{k+1} \leftarrow H^{k+1} \cup H$

12       Generating nodes with features corresponding to elements in $G$ and connecting them to $g^{k-1}$;

13       **if** $k == K-1$ **then**

14         Generating nodes with features corresponding to elements in $H$ and connecting them to $g^{K-1}$.

15       **end**

16     **end**

17  **end**

---

