# OpenReview forum: "Towards Federated Learning of Deep Graph Neural Networks"
_ICLR.cc/2023/Conference — Submitted to ICLR 2023_

### Official Review · Reviewer_53S9 · 2022-10-20

**Confidence:** 4
**Correctness:** 2
**Technical Novelty And Significance:** 3
**Empirical Novelty And Significance:** 2
**Recommendation:** 3

**Clarity, Quality, Novelty And Reproducibility:**

### Clarity
* The exclusion of the father node in the rooted node structure is not convincing. The authors claim that the neighborhood information encoded by the rooted tree of the certain node is the same as the information by the node's ego network. However, if we ignore the farther node, the information of the rooted tree that ignores the father node is different from the information of the ego network that considers the father node.
* The FL dataset generation process is unclear. The output of the used Louvain algorithm is not the number of clients; thus, the authors may merge many different subgraphs, partitioned from the Louvain algorithm, to the subgraphs of the client number (e.g., 10 or 15).

### Quality
* The quality of evaluation is weak. The authors compare only one subgraph FL baseline, namely FedSage+. Also, the efficiency of the proposed method is unclear, while the authors point out that the limitation of existing subgraph FL methods is communication efficiency.

### Novelty
* The proposed method is fundamentally incremental from the existing subgraph FL [2] that proposes to expand the neighborhood of the subgraph, yet the novelty for additionally considering the rooted tree structure is acceptable.

### Reproducibility
* The authors provide the source code, thus the reproducibility is high.

---

[2] Subgraph Federated Learning with Missing Neighbor Generation, NeurIPS 2022.

**Strength And Weaknesses:**

### Strengths
* The concept of the rooted tree structure for encoding the neighboring information for nodes in both the local client and the other clients is interesting.
* The experimental results show that the proposed method outperforms the recent subgraph FL baseline, namely FedSage+.

### Weaknesses
* The proposed subgraph FL method is unrealistic, and not applicable to real-world scenarios. In particular, for the node in the local client, the authors define its neighboring nodes in other clients, and use the information of nodes in other clients based on the edge between the local client and the other clients. However, in privacy-preserving FL, since we cannot share information across clients, we cannot have the information whether two nodes in two different clients have the edge or not.
* Also, the proposed information transmitting process between clients is unrealistic. To share k-hop neighboring information across clients, every client participating in FL should be in the same forwarding process. In other words, when the proposed method shares the information of 2-hop neighborhoods, every client should calculate the 1-hop neighborhood information and share them simultaneously. However, in realistic FL, it is unnatural to share k-hop neighborhood information of all clients simultaneously (i.e., one particular client may update k-hop neighborhood information, meanwhile, the other client may update (k-3)-hop neighborhood information).
* The authors point out that one particular limitation of existing subgraph FL is the communication cost. However, since the proposed method considers k-hop neighborhoods of the root node and consequently all clients should recursively transmit k-hop node information, it seems the proposed one is also highly slow.
* The compared baseline is weak. The authors compares only one subgraph FL baseline, namely FedSage+. I am wondering the authors can further compare the discussed relevant works in the paper: Peng et al., 2021; Yao & Joe-Wong, 2022; Chen et al., 2022.
* There is a related work [1], which tackles the missing neighborhood problem, yet does not rely on neighborhood expansion. This work should be discussed.

---

[1] Personalized Subgraph Federated Learning, arXiv 2022.


**Summary Of The Paper:**

This paper tackles the subgraph-level Federated Learning (FL), where each client has the individual subgraph of the larger global graph. Then, to tackle this task, the authors propose to reconstruct the neighborhood information of the subgraph based on the rooted tree structure. In particular, the rooted tree for the particular node in the local client has the information for its neighboring nodes in the other clients, which is done by aggregating information of nodes in that other clients as well as the nodes in the local client. After that, the authors decode the neighborhood structure of the subgraph with the information in the rooted tree. The experimental results show that the proposed method outperforms the subgraph FL baseline, namely FedSage+.

**Summary Of The Review:**

The biggest concern is that the proposed method is not applicable to real-world scenarios (See first two weaknesses). Also, the evaluation of using one subgraph FL baseline is weak, and the analysis of communication costs is further required. Therefore, I cannot recommend the acceptance.

---

> ### Author Response · Authors · 2022-11-15
> **Response to Reviewer 53S9 (Part 1)**
>
> Thanks for your valuable and constructive feedback. We hope the following clarifications will address your concerns.
>
> --------------
> **Q1: [The assumption of  knowing one-hop neighbors is reasonable]**
>
> **A.** We respectfully disagree with the reviewer. Knowing the one-hop neighbors is common in practice, and such an assumption has also been studied in previous works, such as Yao & Joe-Wong, 2022 and Chen et al., 2022. Consider the scenario where one bank account wants to transfer money to another account registered with a different bank. Although the two accounts are located in different banks, the sender knows the recipient's account, and the recipient knows where the money came from. Treat each account as a node in the money transfer graph and the banks as different clients. Hence, each node knows its one-hop neighbors, whether the neighbors are in the same or different clients. Such a scenario is also common in real-world tasks, such as anti-money laundering, which aims to identify accounts participating in the money laundering process.
>
> -------------------
> **Q2: [The information transmitting process of the method]** *"it is unnatural to share k-hop neighborhood information of all clients simultaneously "*
>
> **A.** Asynchrony in federated learning is an open problem, and further discusses it beyond the scope of our methods. In fact, our methods do not require that all clients be in the same forwarding process. Note that one client recursively collects information for its neighbors. If one client updates its k-hop neighborhood information, it only requires its neighbors to update $(k-1)$-hop neighborhood information. Meanwhile, the client could push forward even if it does not collect all the missing neighborhood information. In the worst cases, the client could push forward if it receives information from one of its neighbors at the cost of losing some information. However, missing part of neighbor information has a minimal effect when the number of neighbors is large.
>
> --------------
> **Q3: [Communication cost]**
>
> We are sorry for the confusion. We have updated Appendix E.3 for a deep analysis of the communication cost of our methods compared with the existing FL frameworks. Specifically, our method only requires the communication cost of $\mathcal{O}(ndK)$, where $n$ is the number of nodes, $d$ is the average degree of all nodes, and  $K$ is the number of layers of a GNN model. Existing FL frameworks require the communication cost of $\mathcal{O}(nd^K)$ (Yao & Joe-Wong, 2022) and $\mathcal{O}(ndT)$ (Chen et al., 2022), where $T>>k$ is the number of iterations. The core insight behind our methods is that we encode the transmitted information first and decode it to reconstruct the neighborhood information by constructing the rooted tree. We also conduct experiments to compare the communication cost of our method with existing frameworks in Appendix E.3. The results show that our methods have the lowest cost in all settings.
>
> ---------------
> **Q4: [Experimental comparison with relevant works]**
>
> **A.** We also compare with Yao & Joe-Wong, 2022 as one of our baselines, named FedAvg-Full, in our experiments. The learning algorithm in Yao & Joe-Wong, 2022 support training GNN models with no information loss, which is in line with the target of our framework in terms of accuracy. However, its communication cost is $\mathcal{O}(nd^K)$. Moreover, it assumes that each node knows any of its k-hop neighbors, while we only assume each node knows its one-hop neighbors.
>
> We did not experiment with Chen et al., 2022 and Peng et al., 2021 as the baseline in terms of accuracy for the following reasons. Chen et al., 2022 proposed a sampling strategy to enable feature sharing among clients. In more detail, it samples missing neighbors and aggregates neighbors' embeddings to generate the next layer's embeddings in each iteration during training. Our framework can easily scale to their methods by sampling nodes on the rooted tree.
> Peng et al., 2021 mainly focus on disease prediction tasks, and the core idea behind it is the same as FedSage+, e.g., generating missing 1-hop neighbors by a node generator.

---

> > ### Author Response · Authors · 2022-11-15
> > **Response to Reviewer 53S9 (Part 2)**
> >
> >
> > **Q5: About the rooted tree** *"The exclusion of the father node in the rooted node structure is not convincing."*
> >
> > **A.** We are sorry for the confusion. The encoding process on the rooted tree includes both the children nodes and the father node. The rooted tree is actually an undirected graph. For any node on the tree, it treats its father node and children nodes as one-hop neighbors and recursively aggregates their information to update its embedding. Meanwhile, the nodes in the $k$-th layer of the rooted tree have the complete neighborhood information of $K+1-k$ hops, and the rooted node at the first layer has the complete $K$-hop neighborhood information. Thus, for a K-layer GNN model, the rooted node has the same embedding as obtained by encoding on its $K$-hop ego-graph.
> >
> > --------
> > **Q6: [The dataset generation process.]** *“The output of the used Louvain algorithm is not the number of clients; thus, the authors may merge many different subgraphs, partitioned from the Louvain algorithm, to the subgraphs of the client number (e.g., 10 or 15).”*
> >
> > **A.** Our dataset generation process follows the work of Zhang et al., 2021. After partitioning a large graph into multiple subgraphs using the Louvain algorithm, we sort the subgroups in decreasing order according to their size. We assign the first M (M is the number of clients) subgraphs to each client. For the remaining subgraphs, we inspect if it is suitable to be assigned to one client. Specifically, suppose the sum of the number of existing nodes in the client and the number of subgraph nodes exceeds the specified value (# nodes / # clients in our experiments). In that case, we discard the subgraph and try a smaller subgraph until the number of nodes does not exceed the value. Looping the above process, we assign all subgraphs to clients, and each client is allocated the same number of nodes.
> >
> > ------------
> > **Q7: [Related work]**
> >
> > **A.** We thank the reviewer for pointing out the related work. We have updated the main text. [1] proposes a personalized federated subgraph learning method, mainly focusing on heterogeneous subgraphs. In more detail, it develops an aggregation method based on similarity among clients. However, it does not consider the missing neighbor information in graph federated learning. Our paper focuses on recovering the neighbor information of nodes in the federated setting and applying it to train deeper GNN models.

---

> > > ### Comment · Reviewer_53S9 · 2022-11-26
> > > **Further reviews**
> > >
> > > Thank you for responding to my reviews.
> > >
> > > ---
> > >
> > > Q1: [The assumption of knowing one-hop neighbors is reasonable]
> > >
> > > I disagree that the assumption of knowing features of one-hop neighbors in the other subgraphs is reasonable. Yes, in the bank transfer scenario, we can know which nodes in the particular bank (i.e., client) send which other nodes in the other bank; however, we cannot know the features of nodes (i.e., transfer history) in the other bank, due to the privacy issue. However, in this paper, the authors access the features of nodes in the other bank, which violates the privacy concern in federated learning.
> > >
> > > ---
> > >
> > > Q2: [The information transmitting process of the method] "it is unnatural to share k-hop neighborhood information of all clients simultaneously"
> > >
> > > I agree that this is a challenging problem; however, the authors would like to consider this as future work, and such a very hard restriction for k-hop information propagation is clearly the weakness of this work.
> > >
> > > ---
> > >
> > > Q6: [The dataset generation process.] “The output of the used Louvain algorithm is not the number of clients; thus, the authors may merge many different subgraphs, partitioned from the Louvain algorithm, to the subgraphs of the client number (e.g., 10 or 15).”
> > >
> > > I have a concern about the data split with the Louvain algorithm. As the authors described, the authors sort the split N subgraphs, and then distribute N into M clients. If N is sufficiently large, which is the usual case in the Louvain algorithm, and some subgraphs have a small number of nodes, then the subgraph that each client has might show weird properties: many nodes are not connected; many nodes come from unrelated split subgraphs.

---

> > > > ### Author Response · Authors · 2022-12-05
> > > > **Followup Response to Reviewer 53S9**
> > > >
> > > > Thank you for your time reading our responses. We want to express our gratitude once more for the useful comments.
> > > >
> > > > ----
> > > > **Q1 [The assumption of knowing one-hop neighbors is reasonable]**   *"in the bank transfer scenario, we can know which nodes in the particular bank (i.e., client) send which other nodes in the other bank; however, we cannot know the features of nodes (i.e., transfer history) in the other bank, due to the privacy issue."*
> > > >
> > > > **A.** We are sorry for the confusion. In our methods, we only assume to know which nodes a node is connected to in other clients, such as knowing the neighbor nodes' unique IDs; we do not assume that a node has access to the features of its neighbor nodes. In more detail, in the feature transmission process, we have passive clients send encodings of neighbor information to the active client. Knowing the IDs of the neighbor nodes only allows the passive clients to know to whom to send the information and the active client knows where the information is sent from. If the passive clients do not send the information to the active client, the active client cannot get the information.
> > > >
> > > > ------
> > > >
> > > > **Q2: [The information transmitting process of the method]**
> > > >
> > > > **A.** We thank the reviewer for understanding the hardness of the problem, and we would like to consider this as future work. The problem appears not only in our methods but also in other methods, such as Chen et al., 2022. Note that our information transmission process is similar to the message-passing scheme. In more detail, both the information transmitting process and the message passing scheme require that the active client (one node) recursively aggregate information from the passive clients (neighbor nodes). Applying the message passing scheme in the federated setting naturally has such a weakness as presented in Chen et al., 2022. However, compared to Chen et al., 2022., which applies the message passing scheme distributedly in each iteration during training, our information transmission process only requires running once. After reconstructing the neighborhood information for each node, our training algorithm applies the message-passing scheme locally; hence the training process is no longer affected by sync issues.
> > > >
> > > > Although Yao & Joe-Wong's work, which proposes to send the missing neighborhood information to the active clients directly, does not have such an issue, it requires the communication cost of $\mathcal{O}(nd^K)$. In contrast, our methods only require the communication cost of $\mathcal{O}(ndK)$. Future works may combine our methods and Yao & Joe-Wong, 2022 to achieve the goal of asynchronous information transmission and low communication cost.
> > > >
> > > > -----
> > > >
> > > > **Q6: [The dataset generation process.]**  *"the subgraph that each client has might show weird properties: many nodes are not connected; many nodes come from unrelated split subgraphs."*
> > > >
> > > > **A.** We are sorry for the confusion. We guess that the "subgraph" may have misunderstood you once. It is more suitable to use "community" to represent the output of the Louvain algorithm. In our experiments, we sort the split N communities and then distribute them to clients. Other than the edges contained in the community, we also preserve the edges between communities assigned to the same client.
> > > >
> > > > Considering the dataset generation process from another perspective, we actually assign nodes in the community to different clients according to the distribution of the community. In more detail, if the community is assigned to one client, we assign nodes in the community to the client. When nodes are distributed on the same client, we will preserve the edges between them regardless of whether they are from the same community or not.

---

> > > > > ### Comment · Reviewer_53S9 · 2022-12-07
> > > > > **Further reviews**
> > > > >
> > > > > Thanks for your further response.
> > > > >
> > > > > ---
> > > > >
> > > > > Q1 **[The assumption of knowing one-hop neighbors is not reasonable]** "in the bank transfer scenario, we can know which nodes in the particular bank (i.e., client) send which other nodes in the other bank; however, we cannot know the features of nodes (i.e., transfer history) in the other bank, due to the privacy issue."
> > > > >
> > > > > As formalized in equations (1), (2), and (3), the suggested methods use the node features of neighboring nodes. Also, the considered datasets, such as Cora, MSAcademic, and DBLP, have node features. In the previous response, the authors said "we only assume to know which nodes a node is connected to in other clients, such as knowing the neighbor nodes' unique IDs", which is also different from the formalizations in equations (1), (2), and (3). If the proposed method only shares node IDs, which information is shared between clients?
> > > > >
> > > > > ---
> > > > >
> > > > > Q2 **[The information transmitting process of the method]** "it is unnatural to share k-hop neighborhood information of all clients simultaneously"
> > > > >
> > > > > The authors suggest to leave it as future work, however, I still believe that sharing k-hop information across all clients simultaneously is not realistic in federated learning, which is a clear weakness of this work.
> > > > >
> > > > > ---
> > > > >
> > > > > Q6: **[The dataset generation process]** "the subgraph that each client has might show weird properties: many nodes are not connected; many nodes come from unrelated split subgraphs."
> > > > >
> > > > > The explanations are still unclear. Could you share the number of split subgraphs (M) after the Louvain algorithm, and the number of clients (N) across all datasets? I think M is larger than N since the authors use the small N (e.g., 5) while M is not controllable, and, if so, I wonder how to merge a few split subgraphs into one client.

---

> > > > > > ### Author Response · Authors · 2022-12-10
> > > > > > **Followup Response to Reviewer 53S9**
> > > > > >
> > > > > > Thank you for your further reviews.
> > > > > >
> > > > > > -----------------
> > > > > >
> > > > > > **Q1 [The assumption of knowing one-hop neighbors is reasonable]** *"If the proposed method only shares node IDs, which information is shared between clients?"*
> > > > > >
> > > > > > The reviewers may have misunderstood the part of our previous response regarding the assumptions, and we would like to clarify. In our follow-up responses, we said, "we only assume to know which nodes a node is connected to in other clients, such as knowing the neighbor nodes' unique IDs". We clarify that we use the word "know" here to indicate that the node has the IDs of its neighbor nodes. For example, the client can maintain a database locally, which records the IDs of the neighbor nodes of each node located on the client. Hence, the node IDs are not transmitted among clients.
> > > > > >
> > > > > > We then said, "we do not assume that a node has access to the features of its neighbor nodes". The meaning of this sentence is that the node does not have features of nodes in the other clients. For instance, a node $i$ located in client $A$ may have neighbor nodes located in client $B$. $B$ maintains a database that records the features of nodes located on it (including the neighbor nodes of $i$). Client $A$ has no access to the database. The GNN models in our work consider the features of nodes. Hence, to tackle the problem of missing neighborhood information, we design a neighbor information transmission protocol (Protocol 1 in Appendix F) to transmit encodings of feature information (line 6 and line 15 in Protocol 1) between clients. The protocol allows clients to reconstruct the missing neighborhood information (including structure- and feature-based information) for training GNN models.
> > > > > >
> > > > > > Note that transmitting neighbor information between clients also exists in previous works, e.g., Chen et al., 2022, and Yao & Joe-Wong, 2022. In more detail,  Chen et al., 2022. propose to aggregate neighbors' embeddings to generate the next layer's embeddings in each iteration during training. In the federated setting, it requires passive clients to send embeddings of neighbor nodes to the active client. Yao & Joe-Wong, 2022 propose sending the missing feature information to the active clients directly. The main difference (advantage) of our protocol is that it only requires the communication cost of $\mathcal{O}(ndK)$, while Chen et al., 2022 requires $\mathcal{O}(ndT)$ and Yao & Joe-Wong, 2022 requires $\mathcal{O}(nd^K)$.
> > > > > >
> > > > > > ------------------
> > > > > > **Q6: [The dataset generation process]** *"I wonder how to merge a few split subgraphs into one client."*
> > > > > >
> > > > > > The number of split subgraphs (M) after the Louvain algorithm for Cora, MSAcademic, and DBLP is 106, 27, and 627. The number of split subgraphs of the DBLP dataset is large because the original graph contains many independent subgraphs of size less than 10, except for one large subgraph of size 16191. The number of clients (N) is deterministic, and we set 5, 10, and 15 in our experiments.
> > > > > >
> > > > > > Our merging procedure follows the work of Zhang et al., 2021 and has been presented in our first response. We rewrite it more clearly and present it here. We first partition a large graph into M subgraphs using the Louvain algorithm. We then sort subgraphs in descending order according to their size and assign the first N (N is the number of clients) subgraphs to N clients. For the next (N+1)-th subgraph, we inspect if it is suitable to be assigned to the first client. Specifically, suppose the sum of the number of existing nodes in the client and the number of nodes in the subgraph is lower than the specified value (the number of nodes of the original graph / M in our experiments). In that case, we assign the subgraph to the client, or we try a subgraph with a smaller size. Following the above process, we merge several subgraphs to the first client, where the number of nodes in the client is the number of nodes of the original graph / M. We then generate the second client following the same principle using the remaining unassigned subgraphs. Looping the above process, we merge all N subgraphs into N clients, and each client has (basically) the same number of nodes.

---

### Official Review · Reviewer_YSgt · 2022-10-23

**Confidence:** 2
**Clarity, Quality, Novelty And Reproducibility:** <included in the comments above.>
**Correctness:** 3
**Technical Novelty And Significance:** 3
**Empirical Novelty And Significance:** 3
**Recommendation:** 6

**Strength And Weaknesses:**

Pros:
1. The related works are well explained with their strengths and drawbacks nicely pointed out.
2. The construction of the rooted tree and then the procedure to do k-hop neighborhood reconstruction seems correct, though I have not verified the proofs given in the Appendix.
3. The set of experiments chosen are well thought-off, although I do share certain concerns with the size of the datasets.

Comments:
1. (Fig error) In fig.1(b) the node below (i) should be (m2). Also, please number all the equations.
2. Can you please comment on the scalability of the method? The experimental datasets are very small sized graphs. In general, it will be of interest to the readers to get a sense of the time required for rooted tree construction, aggregation steps etc.?
3. How does the memory requirement scale while creating the rooted tree for every node? Do you foresee issues with larger graphs in this regard?
4. Any best practice on choosing the value of K (layers)  ?
5. How can one do BFS vs DFS tradeoff while aggregating neighborhood information using the rooted tree graph structure?

I think this work is novel, clearly presented and interesting. Although, I am not entirely confident about the scalability of their approach and have listed some of my concerns in the comments above. I will be happy to reconsider my ratings after replies from the authors.


**Summary Of The Paper:**

The authors propose a federated learning model for learning deep GNNs to solve node-level prediction tasks. Their idea is based on reconstructing the neighborhood information of nodes that accounts for neighborhood graph structure as well as their features in a principled manner. For aggregating neighborhood structure information, they proposed a rooted tree graph approach, whose node embedding is the same as the node’s eco-graph. Reconstructing feature-based information for a given node is handled by an encoder-decoder framework with some information loss. Their experiments show improvement over SOTA.

Update: The authors have addressed my questions to certain extent and I've updated my ratings to reflect the same. Although, due to my lack of breadth in federated learning, the confidence in my assessment is on the lower end.

**Summary Of The Review:**

I think this work is novel, clearly presented and interesting. Although, I am not entirely confident about the scalability of their approach and have listed some of my concerns in the comments above. I will be happy to reconsider my ratings after replies from the authors.

---

> ### Author Response · Authors · 2022-11-15
> **Response to Reviewer YSgt (Part 1)**
>
>
> Thanks for your valuable and constructive feedback. We hope the following clarifications will address your concerns.
>
> ---------------
> **Q1. [Typo in Figure 1 and add equation numbers]**
>
> **A.** We thank the reviewer for pointing out the typo and for the advice. We have updated the main text.
>
> ---------------
> **Q2. [Scalability of the method]** *"In general, it will be of interest to the readers to get a sense of the time required for rooted tree construction, aggregation steps etc.?"*
>
> **A.** The process of building a rooted tree is quite simple. It mainly consists of a sequence of predicting processes of the decoder $\phi$.  Algorithm 2 presents the pseudo-code of the process for one tree. For each vector $h \in H^k$, the decoder takes it as input and outputs $G$ and $H$ (line 9 in Algorithm 2). The $K+1$-th layer of the tree is constructed by generating nodes with features corresponding to elements in $G$ and connecting them to node $g$ corresponding to $h$.
>
> Furthermore, in practice, the vectors in the set $H^k$ can be fed to the decoder simultaneously. Meanwhile, we can even input all $H^1$s (of different root nodes) into the decoder to construct the rooted trees simultaneously. To that end, building rooted trees of $K$ layers for all nodes requires the decoder to predict only $K-1$ times. To generate a node, we simply append the generated feature vector to the array that stores all node features. To generate an edge, we append the node indexes to a two-dimensional array, where the first dimension stores the source node index and the second dimension stores the target node index. In summary, the tree-building process is efficient.
>
> After building the rooted trees for all nodes with missing neighborhood information, we merge the trees into the local graph (with the rooted node as an anchor) to get a complete graph. We apply the message-passing scheme on the complete graph to aggregate information from the original local graph and the trees. Such a scheme is widely used in existing GNN models.
>
> We also add experiments to evaluate the efficiency of our methods. We vary the number of layers and report both the tree-building time and training time. The results are shown below and we have updated them in Appendix E.4. The results show we can construct rooted trees for all nodes within a few seconds.
>
> | Cora                   | 3      | 4      | 5      | 6      | 7      | 8      | 9      |
> | ---------------------- | ------ | ------ | ------ | ------ | ------ | ------ | ------ |
> | Tree-building time (s) | 0.0406 | 0.0584 | 0.0870 | 0.1396 | 0.2558 | 0.5211 | 1.1944 |
> | Training time (s)      | 0.2644 | 0.2898 | 0.3074 | 0.3258 | 0.3414 | 0.3592 | 0.3965 |
>
> | DBLP                   | 3      | 4      | 5      | 6      | 7      | 8      | 9      |
> | ---------------------- | ------ | ------ | ------ | ------ | ------ | ------ | ------ |
> | Tree-building time (s) | 0.1770 | 0.2879 | 0.4529 | 0.7361 | 1.4565 | 3.1793 | 7.6464 |
> | Training time (s)      | 0.5499 | 0.5941 | 0.6246 | 0.6610 | 0.6934 | 0.9186 | 1.7850 |
>
> | MSAcademic             | 3      | 4      | 5      | 6       |
> | ---------------------- | ------ | ------ | ------ | ------- |
> | Tree-building time (s) | 0.4068 | 1.3076 | 3.7498 | 14.4787 |
> | Training time (s)      | 0.5755 | 0.7562 | 1.7637 | 6.7981  |
>
> ----------------
> **Q3. [The memory requirement of the method]**
>
> **A.** We only conduct rooted trees for nodes that have one-hop neighbor nodes located in other clients. Meanwhile, one advantage of our approach is that we only reconstruct **missing** neighborhood information, hence could significantly reduce the required memory. Take Figure1 (c) as an example. If the only node (m1) is missing, then we only construct the (m1) branch in the second layer along with the children nodes in that branch, which saves 2/3 of memory compared with reconstructing the whole rooted tree.
>
> Furthermore, recall that we merge the trees into the local graph (with the rooted node as an anchor) to get a complete graph. The complete graph is stored in the format the same as the original graph (using the PyTorch-geometric package in our experiments), and perform training on it follows the same training paradigm as ordinary graph learning. Overall, our methods can easily scale to large graph data.

---

> > ### Author Response · Authors · 2022-11-15
> > **Response to Reviewer YSgt (Part 2)**
> >
> > **Q4. [Determing the number of layers (K)]**
> >
> > **A.** Determining the number of layers ($K$) for one dataset is not easy, even in a centralized setting. Existing work [1] empirically shows that when nodes are more densely connected, their representations will become indistinguishable by applying a smaller number of propagation iterations, which connects the graph topology information and convergence speed. In practice, each client can locally train GNN models with different layers to choose the $K$ used for federated learning. Moreover, when the data of each client is non-i.i.d, notice that the number of layers has a similar impact on our method and FedAvg (as presented in Figure 3); all clients can pre-train GNN models with the FedAvg algorithm to choose $K$.
> >
> > Meanwhile, although $K$ is hard to determine, our method for reconstructing the neighborhood can easily scale to different layers. Specifically, after reconstructing the $k$-th neighborhood, if $k+1$-th neighborhood is needed, it is only required to consider transmitting $k$-hop neighborhood information from neighbor nodes instead of transmitting information from scratch.
> >
> >
> > ---------------
> > **Q5. [The aggregation process on the rooted tree]**
> >
> > **A.** The aggregation process on the rooted tree is neither BFS nor DFS. In our methods, we merge all trees into the local graph to get a complete graph. We aggregate the complete graph with the message-passing scheme, which has been widely used in GNN models.
> >
> > [1] Towards Deeper Graph Neural Networks, KDD'20

---

> > > ### Comment · Reviewer_YSgt · 2022-12-05
> > > **Rating updated**
> > >
> > > Hello authors, I've updated my ratings based on your response to my questions.

---

> > > > ### Author Response · Authors · 2022-12-05
> > > > **Sincerely Thankful for Raising the Score**
> > > >
> > > > We appreciate the reviewer for raising the score to 6! Thanks for the valuable comments and suggestions!

---

### Official Review · Reviewer_SQ7X · 2022-10-31

**Confidence:** 4
**Correctness:** 3
**Technical Novelty And Significance:** 3
**Empirical Novelty And Significance:** 3
**Recommendation:** 5

**Clarity, Quality, Novelty And Reproducibility:**

The paper is mostly clear, but some parts of the paper needs further explanations.

**Strength And Weaknesses:**

The authors claim that the node embedding obtained by encoding on the rooted tree is the same as that obtained by encoding on the induced subgraph surrounding the node. This is captured in proposition 1. The proof is included in Appendix A. One suggestion is, given the importance of the proposition, it is necessary to include at least a sketch version of it in the main doc.

I don’t quite get the challenge of the proposed setting. This paper assumes that every node knows its one-hop neighbors, no matter whether the neighbors are in the same host or different host. Is that a reasonable assumption? If yes, the information can of course help to identify k-hop neighborhoods. Is that a challenging setting?

The authors criticize Yao & Joe-Wong, 2022 for making the assumption that the weight matrix is calculated in advance. Which weight matrix? There are no detailed explanations. Given a similar idea were proposed in that paper, it is important for the authors to explain the underlying reasoning.

Building a root tree for each node is expensive. The authors do not seem to have experimental evaluations to evaluate the efficiency of the proposed method. What were the criteria of experiments for comparing different methods? The same amount of running time, or same number of iterations?

**Summary Of The Paper:**

The authors develop a method for federated learning of deep GNNs. The main idea is to reconstruct the neighborhood information of nodes using a graph structured named rooted tree, and use the rooted tree for encoding neighborhood information.

**Summary Of The Review:**

The paper proposes to build rooted tree for each node in a GNN for federated learning. The approach in general makes sense. However, the practicality of the proposed method might be somewhat limited.

---

> ### Author Response · Authors · 2022-11-15
> **Response to Reviewer SQ7X (Part 1)**
>
> Thanks for your valuable and constructive feedback. We hope the following clarifications will address your concerns.
>
> ---------------
> **Q1: [Proof sketch of Proposition 1]** *"One suggestion is, given the importance of the proposition, it is necessary to include at least a sketch version of it in the main doc."*
>
> **A.** We appreciate the valuable suggestions and have updated the main text. We also present it below. Note that the rooted tree is actually an undirected graph. For any node on the tree, it treats its father node and children nodes as one-hop neighbors and recursively aggregates their information to update its embedding. Meanwhile, the nodes in the $k$-th layer of the rooted tree have the complete neighborhood information of $K+1-k$ hops, and the rooted node at the first layer has the complete $K$-hop neighborhood information. Thus, for a $K$-layer GNN model, the rooted node has the same embedding as obtained by encoding on its $K$-hop ego-graph.
>
> -------------
> **Q2: [The assumption of  knowing one-hop neighbors is reasonable]**
>
> **A.** Knowing the one-hop neighbors is common in practice, and such an assumption has also been studied in previous works, such as Yao & Joe-Wong, 2022 and Chen et al., 2022. Consider the scenario where one bank account wants to transfer money to another account registered with a different bank. Although the two accounts are located in different banks, the sender knows the recipient's account, and the recipient knows where the money came from. Treat each account as a node in the money transfer graph and the banks as different clients. Hence, each node knows its one-hop neighbors, whether the neighbors are in the same or different clients. Such a scenario is also common in real-world tasks, such as anti-money laundering, which aims to identify accounts participating in the money laundering process. In addition, Yao & Joe-Wong, 2022 also have the same assumption, and they even assume that one node knows its $k$-hop neighbors.
>
> Given the one-hop neighbors, constructing the $k$-hop neighborhood information is still challenging. One natural approach to finding k-hop neighbors for one node is applying DFS or BFS. However, it would leak the privacy of the intermediate node (e.g., knowing $k$-hops neighbors and ($k-1$)-hop neighbors could reveal the privacy of ($k$-1)-hop neighbors since we know that the $k$-hops neighbors are their one-hop neighbors).
>
> Meanwhile, transferring neighbor information between clients is by no means trivial, even if k-hop neighbors are known. In more detail, the communication cost requires $\mathcal{O}(nd^K)$, where $n$ is the number of nodes, $d$ is the average degrees, and $K$ is the number of GNN layers (as presented in Yao & Joe-Wong, 2022). However, in our paper, we have one node recursively collect information from its one-hop neighbors. Hence, our methods only require that one node knows its one-hop neighbors, and the communication cost is $\mathcal{O}(ndK)$.
>
> -------------
> **Q3: [The weight matrix in Yao & Joe-Wong, 2022]**
>
> **A.** We are sorry for the confusion. The weight matrix in Yao & Joe-Wong, 2022 refers to the weight matrix in the GCN model. It has the expression of  $\hat{A}=\tilde{D}^{-\frac{1}{2}} \tilde{A} \tilde{D}^{-\frac{1}{2}}$, where $\tilde{A}=A+I_N$ is the adjacency matrix of the undirected graph with added self-connections and $\tilde{D}_{ii} =\sum_j\tilde{A}_\{ij\}$. In order to calculate $ \hat{A} $, not only neighbors of each node but also the degree of neighbors are required, where the latter one would leak the privacy of neighbor nodes. Hence, it isn't easy to achieve in real-world scenarios.
>
> Whereas in our paper, we do not need to calculate the weight matrix in advance. We manipulate the edge weight of the rooted tree such that it also supports training GCN models. The details are presented in Appendix A.

---

> > ### Author Response · Authors · 2022-11-15
> > **Response to Reviewer SQ7X (Part 2)**
> >
> > **Q4: [The efficiency of the tree-building process]**
> >
> > **A.** The process of building a rooted tree is quite simple. It mainly consists of a sequence of predicting processes of the decoder $\phi$.  Algorithm 2 presents the pseudo-code of the process for one tree. For each vector $h \in H^k$, the decoder takes it as input and outputs $G$ and $H$ (line 9 in Algorithm 2). The $K+1$-th layer of the tree is constructed by generating nodes with features corresponding to elements in $G$ and connecting them to node $g$ corresponding to $h$.
> >
> > Furthermore, in practice, the vectors in the set $H^k$ can be fed to the decoder simultaneously. Meanwhile, we can even input all $H^1$s (of different root nodes) into the decoder to construct the rooted trees simultaneously. To that end, building rooted trees of $K$ layers for all nodes requires the decoder to predict only $K-1$ times. To generate a node, we simply append the generated feature vector to the array that stores all node features. To generate an edge, we append the node indexes to a two-dimensional array, where the first dimension stores the source node index and the second dimension stores the target node index. In summary, the tree-building process is efficient.
> >
> > We also add experiments to evaluate the efficiency of our methods. We vary the number of clients and report both the tree-building time and training time. Partitioning more clients makes more nodes missing one-hop neighbor information and thus takes more time to build trees. We conduct experiments on a server equipped with an A100 GPU. The results are shown below, and we have updated them in Appendix E.4. The results show we can construct rooted trees for all nodes within a few seconds.
> >
> > | Cora                   | 3      | 4      | 5      | 6      | 7      | 8      | 9      |
> > | ---------------------- | ------ | ------ | ------ | ------ | ------ | ------ | ------ |
> > | Tree-building time (s) | 0.0406 | 0.0584 | 0.0870 | 0.1396 | 0.2558 | 0.5211 | 1.1944 |
> > | Training time (s)      | 0.2644 | 0.2898 | 0.3074 | 0.3258 | 0.3414 | 0.3592 | 0.3965 |
> >
> > | DBLP                   | 3      | 4      | 5      | 6      | 7      | 8      | 9      |
> > | ---------------------- | ------ | ------ | ------ | ------ | ------ | ------ | ------ |
> > | Tree-building time (s) | 0.1770 | 0.2879 | 0.4529 | 0.7361 | 1.4565 | 3.1793 | 7.6464 |
> > | Training time (s)      | 0.5499 | 0.5941 | 0.6246 | 0.6610 | 0.6934 | 0.9186 | 1.7850 |
> >
> > | MSAcademic             | 3      | 4      | 5      | 6       |
> > | ---------------------- | ------ | ------ | ------ | ------- |
> > | Tree-building time (s) | 0.4068 | 1.3076 | 3.7498 | 14.4787 |
> > | Training time (s)      | 0.5755 | 0.7562 | 1.7637 | 6.7981  |
> >
> > ---------------
> > **Q5:  [The criteria of experiments for comparing different methods]**
> >
> > **A.** We are sorry for the confusion. We report the best performance of different models, with the hyper-parameters chosen from Table 5 and Table 6 in Appendix D.

---

> > > ### Author Response · Authors · 2022-12-06
> > > **Message to reviewer SQ7X**
> > >
> > > Dear Reviewer SQ7X,
> > >
> > > We'd like to express our gratitude once more for your constructive suggestions, which resulted in interesting revision updates. We've responded to each of your questions. Hopefully, you'll find that they adequately address your concerns. Additionally, we'd like to know if you have any additional questions or require clarification before the rebuttal phase concludes. We would be delighted to address them in the revision.
> > >
> > > Best wishes,
> > >
> > > Authors of Paper

---

### Comment · Area_Chair_aRAN · 2022-11-22
**Please respond as soon as possible if you still have questions on the paper.**

Please respond as soon as possible if you still have questions on the paper.

---

> ### Comment · Area_Chair_aRAN · 2022-11-29
> **Please respond to the authors by Nov. 30**
>
> Please indicate whether the authors' rebuttal addresses your concerns.
>
> If you still have questions, please ask as soon as possible.

---

> > ### Comment · Area_Chair_aRAN · 2022-12-05
> > **Zoom Meeting**
> >
> > For all reviewers, which have not responded to the authors, I will have to ask you to meet via Zoom. If you want to avoid such an additional step, please respond by Dec. 5.

---

> > > ### Comment · Reviewer_YSgt · 2022-12-05
> > > **Responded to the authors**
> > >
> > > Hello area chair,
> > >
> > > I've responded to the authors accordingly.
> > >
> > > Thanks,
> > > Reviewer YSgt

---

### Decision · Program_Chairs · 2023-01-20

**Decision:**

Reject

**Justification For Why Not Higher Score:**

NA

**Justification For Why Not Lower Score:**

NA

**Metareview: Summary, Strengths And Weaknesses:**

This paper presents a method for federated learning (FL) at the subgraph level, in which each client has a separate subgraph of a larger global graph. The method reconstructs the neighborhood information of the subgraph using a rooted tree structure and outperforms the FedSage+ baseline according to experimental results.

The reviewers raised some concerns on the practicality of the proposed method. These concerns are critical, as they are questioning the key assumptions of the paper. The concerns remained after discussions. The paper will need some major revision to better motivate their method and justify those assumptions.



**Summary Of Ac-Reviewer Meeting:**

NA